# A patient-based iPSC-derived hepatocyte model of alcohol-associated cirrhosis reveals bioenergetic insights into disease pathogenesis

Bani Mukhopadhyay [1], Cheryl Marietta[1], Pei-Hong Shen[1], Abdul Oiseni[2], Faridoddin Mirshahi[2], Maria Mazzu[1], Colin Hodgkinson[1], Eli Winkler[1], Qiaoping Yuan [1], Daniel Miranda[3], George Kunos[4], Arun J. Sanyal [2] & David Goldman [1,5] ✉

Only ~20% of heavy drinkers develop alcohol cirrhosis (AC). While differences in metabolism, inflammation, signaling, microbiome signatures and genetic variations have been tied to the pathogenesis of AC, the key underlying mechanisms for this interindividual variability, remain to be fully elucidated. Induced pluripotent stem cell-derived hepatocytes (iHLCs) from patients with AC and healthy controls differ transcriptomically, bioenergetically and histologically. They include a greater number of lipid droplets (LDs) and LD-associated mitochondria compared to control cells. These pre-pathologic indicators are effectively reversed by Aramchol, an inhibitor of stearoyl-CoA desaturase. Bioenergetically, AC iHLCs have lower spare capacity, slower ATP production and their mitochondrial fuel flexibility towards fatty acids and glutamate is weakened. MARC1 and PNPLA3, genes implicated by GWAS in alcohol cirrhosis, show to correlate with lipid droplet-associated and mitochondria-mediated oxidative damage in AC iHLCs. Knockdown of PNPLA3 expression exacerbates mitochondrial deficits and leads to lipid droplets alterations. These findings suggest that differences in mitochondrial bioenergetics and lipid droplet formation are intrinsic to AC hepatocytes and can play a role in its pathogenesis.

In the past year, some 29.5 million Americans ages 12 and older had alcohol use disorder (AUD)[1]. In the US, 1.8% of U.S. adults (4.5 million people) had liver disease of some nature[2] and in 2021, of 100,530 deaths due to liver disease, 47.4% involved alcoholic liver disease (ALD), including alcoholic cirrhosis (AC) and alcoholic hepatitis (AH), both which are trending upward in prevalence, as is AUD itself. Further, the incidence of alcohol-associated deaths have rapidly accelerated during the COVID-19 pandemic, while the ongoing global epidemic of obesity presage future increases in the mortality and morbidity associated with ALD, as well as

[1]Laboratory of Neurogenetics, National Institute on Alcohol Abuse and Alcoholism, NIH, Bethesda, MD 20892, USA. [2]Division of Gastroenterology, Hepatology, and Nutrition, Department of Internal Medicine, Virginia Commonwealth University School of Medicine, Richmond, VA 23298, USA. [3]Aivia Machine Learning Team, Leica Microsystems, Inc, Deerfield, IL, USA. [4]Laboratory of Physiologic Studies, National Institute on Alcohol Abuse and Alcoholism, NIH, Bethesda, MD 20892, USA. [5]Office of the Clinical Director, National Institute on Alcohol Abuse and Alcoholism, NIH, Bethesda, MD 20892, USA. ✉e-mail: davidgoldman@mail.nih.gov

non-alcoholic steatohepatitis (NASH) and non-alcoholic fatty liver disease (NAFLD).

Each of these liver diseases just mentioned represent a distinct clinical entity their mechanisms partially overlap and synergize, with strong histopathologic parallels between AC and NAFLD (Sanyal et al., 2021). Genome-wide association studies (GWASs) have implicated genes common to AC and NAFLD, and, to specify one physiologic overlap, the combination of alcohol and a high-calorie diet is more likely to lead to liver disease than is either of these provocations alone[3,4]. Although the manifestations of ALD are diverse, steatosis, steatohepatitis and fibrosis usually follow sequentially. In its early phases, ALD is reversible, but both AC and AH can lead to end-stage liver disease, for which transplantation is the only long-term treatment. Approximately 8000 liver transplants are performed annually in the US but as many as twice that number are on waiting lists and face high mortality rates, with overall pre-transplant mortality in 2019 being 12.3 per 100 waiting list-years[5]. In one-third of liver transplant cases alcohol is thought to have been the primary cause of liver failure.

Animal models of ALD and NAFLD have led to new insights into pathogenetic mechanisms for each of these conditions, with potential new therapeutics identified[6]. However, interspecies differences in bioenergetics and genetics limit the translational relevance of these complementary animal models. Furthermore, genetic factors unique to individuals and their blood relatives, or certain populations, as identified by GWAS, suggest the necessity of controlled human experimental models, especially cell-based models. Hepatocytes are the primary parenchymal cells of the liver, constituting approximately 80% of total hepatic mass. Hepatocytes perform key roles in lipid, glutamate and carbohydrate metabolism; the synthesis of major blood components, such as albumin and transferrin; the detoxification of exogenous and endogenously synthesized molecules; in glycogen and fat energy storage; and the distribution of energy molecules packaged in chylomicrons[7]. As part of their role in detoxification, hepatocytes metabolize alcohol, thereby generating metabolites that are both directly toxic (e.g., acetaldehyde and free radicals) and indirectly toxic (e.g., acetate when in excess) and that contribute to inflammation and fibrosis during the course of ALD[8,9].

To better understand the role of hepatocytes in ALD, we generated homogeneous induced pluripotent stem cell-derived hepatocytes (iHLCs) from individuals with AC and from healthy controls, hypothesizing that AC iHLCs would differ bioenergetically and transcriptomically. and perhaps also histologically, from healthy (H) iHLCs. We turned to the use of iPSC-derived iHLCs, rather than relying on primary hepatocytes because of the many technical and practical challenges of deriving primary hepatocytes from patients. Although iHLCs have recently been used in many studies, including in regenerative medicine[10,11], no data on AC iHLCs and the underlying pathomechanisms that may be revealed through their study have been reported to date.

GWAS of AC has implicated several genes, including *PNPLA3* and *HNF1*, that are strongly, if not uniquely, expressed in hepatocytes and that are critical for hepatocyte function. Three of the top five statistical hits in a meta-analytic GWAS implicated genes are involved in lipid droplet formation[12]. This observation led us to hypothesize that AC iHLCs would be intrinsically different than H iHLCs and could reveal aspects of physiological vulnerability critical to AC progression, even as studies of other cells, such as Kupffer cells, hepatic stellate cells and cholangiocytes, or multi-cellular organoids, might be informative for other aspects of innate vulnerability to AC. Furthermore, we recognized that while a cellular, non-organoid-based iHLC model would not capture extrahepatic mechanisms, including the effects of the gut microbiome, AC iHLCs might show functional differences attributable to genes thus far implicated by GWAS, including *PNPLA3, MARC1, FAF2,*

*HSD17B13, HNF1A* and *SERPINA1*[12,13]. Finally, despite progress being made via GWAS, most of the inter-individual variance in vulnerability to ALD remains unexplained. By comparing iHLCs derived from cases and controls we could capture unknown components of the AC genetic background underlying susceptibility to progression from healthy hepatic function to steatosis and AC.

## Results

### Generation of homogeneous iHLCs from peripheral blood cells of healthy controls and patients with AC

The timeline for the generation of iPSCs processed as case/control pairs within batches was consistent across batches of blood samples and between cases and controls (Fig. 1A). The differentiation, growth and morphology of iPSC clones were similar at key time points (Fig. 1B) and by different criteria, including the initial gain and later loss of SSEA4 expression, indicative of pluripotency (Fig. 1C). Highly homogeneous (>90%) populations of iHLC's were thereby obtained from blood samples of healthy individuals and patients with AC ($n = 6$ for each group) whose clinical characteristics are shown in Table 1. The maturity of iHLCs as hepatocytes was determined by intracellular staining of albumin and HNF4 (Fig. 1D). Additional markers of hepatocyte maturity were determined by real-time PCR estimation of transcript of the genes *AFP, CYP3A, ATT, ABCC2, MAOB* and *UGT1A1* from isolated RNA from Day 1 and Day 21 of iPSC-to-iHLC differentiation. The expression levels of all these hepatocyte -characteristic transcript were significantly greater at Day21 compared to Day 1 of the differentiation process (Fig. S1A). Likewise, a similar increase in AFP protein levels was validated using ELISA (Fig. S1B). Robust expression of the hepatocyte surface marker E cadherin and cuboidal morphology were also observed, whereas iHLCs lacked expression of OCT4, an iPSC marker (Fig. 1E, F). We also individually assayed the mRNA expression of several other genes characteristic of mature hepatocytes; namely, *CYP1A2, CYP2C9, PROX1* and *TBX3*, and all were significantly higher in iHLCs compared to iPSCs (Fig. 1G).

The functional aspects of iHLCs were determined at Days 1 and 21 and compared with primary human hepatocytes. We found that secretion of urea, the storage of glycogen and CYP450 activity, hallmarks of mature hepatocyte function, were significantly greater at Day 21 compared to Day 1, but they were all significantly lower than in primary hepatocytes (Fig. S2A–C). It was also important to determine whether the cells underwent cellular stress at the end of maturation. We looked at two different cellular stress and cell death markers, PARP activity and DNA fragmentation, and both were not elevated at Day 21 compared to Day 1 (Supplementary Fig. S3).

Together, these expression and functional data support previous observation that iHLCs represent an in vitro cellular model of mature hepatocyte physiology and bioenergetics iHLCS potentially being most representative of "young" hepatocytes.

### Patient-derived AC iHLCs accumulate more and larger lipid droplets compared to H iHLCs

LDs are fat-storage organelles that serve as readily accessible reservoirs of high-energy substrates, including fatty acids, that are used for fuel-generating β-oxidation within mitochondria. Although both AC iHLCs and H iHLCs were equivalent in cuboidal morphology and expression of hepatocyte markers, the number of LDs was 4.1-fold greater ($P = 0.002$) in AC iHLCs than H iHLCs. (Fig. 2A, B), as determined by Nile Red staining. A second LD stain, Adipo Red, yielded a similar finding (fold difference 2.9, $P = 0.0014$) (Fig. 2C) and excess of LDs in AC iHLCs compared to H iHLCs was also verified using the neutral lipid stain HCS LipidTOX™ Green (Fig. 2D).

Next, we probed two critical parameters; namely, de novo lipogenesis and fatty acid uptake, that might explain enhanced LD

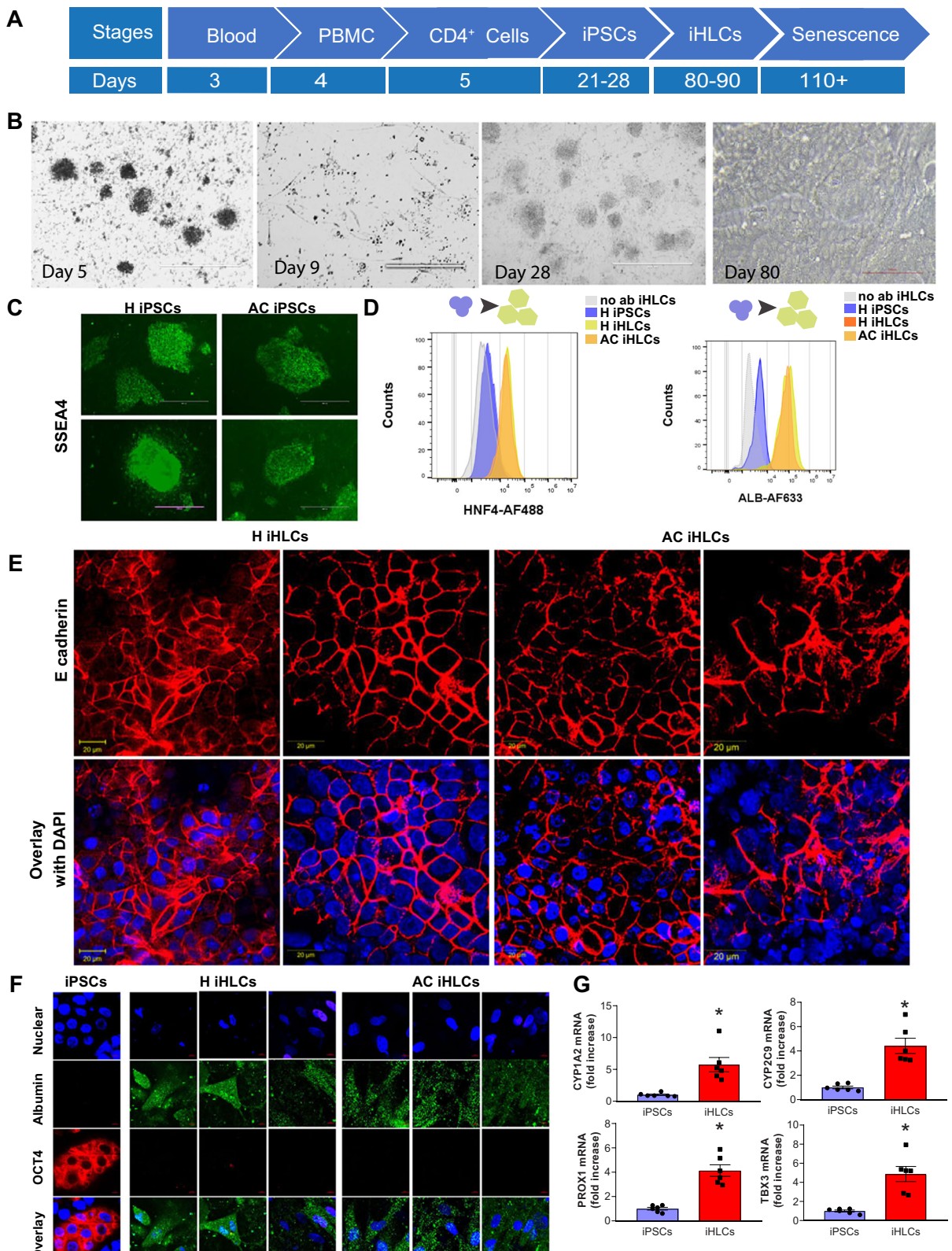

formation in AC iHLCs. Compared to H iHLCs, AC iHLCs exhibited a nonsignificant trend of 68% greater de novo lipogenesis (Fig. 2E) and a significantly 47% greater fatty acid uptake ($P = 0.0029$; Fig. 2F). Furthermore, the triglyceride content of AC iHLC's was 2.4-fold higher ($P = 0.003$) and phosphatidylcholine was 4.4-fold ($P < 0.0001$) higher compared to H iHLCs (Fig. 2G, H). Confocal imaging of iHLC's doubly stained for neutral lipid and mitochondria revealed that LDs in AC iHLCs tended to be more closely associated with mitochondria than those in H iHLCs (Fig. 2I, J).

## AC iHLCs have unique transcriptomic signatures for cellular bioenergetics

By RNA-sequencing of six AC iHLC and four H iHLC lines, we identified 343 differentially expressed genes (DEGs) ($P < 0.05$). Critically, cell line-

**Fig. 1 | Generation and analysis of iPSC-derived iHLCs from AC and healthy controls. A** A schematic diagram illustrating the timeline for the generation of iHLCs from peripheral blood T lymphocytes. Scale bar 400 μm. **B** Phase-contrast imaging at day 5, 9, 29 and 80 after initiation from blood samples showing progression to cuboid iHLC morphology. Scale bar 400 μm. **C** De-differentiation of B lymphocytes to iPSCs and differentiation to iHLCs shown by staining of representative healthy (H iHLC) and alcohol cirrhosis (AC iHLC) lines for SSEA4, an indicator of pluripotency. Scale bar 400 μm. **D** Flow cytometry analyses with HNF4α and albumin staining showing differentiation to iHLCs from iPSCs **E** Representative images showing iHLC staining of E-cadherin, along with a nuclear stain (DAPI). Scale bar 20 μm. **F** Representative images illustrating maturity and homogeneity of iHLCs using antibodies to albumin versus the iPSC-specific marker OCT4. Scale bar 10 μm **G** Quantitation of hepatocyte-specific transcripts CYP1A2, CYP2C9, PROX1, and TBX3 by real-time PCR and comparison of iHLCs with iPSCs ($n = 6$, *$P = 0.0018$, $0.0003$, $<0.0001$ and $0.0007$ by unpaired two-tailed t-test for CYP1A2, CYP2C9, PROX1, and TBX3 respectively). All data were shown as mean ± SEM. All experiments were replicated three times or more.

to-cell line correlations of transcript expression were remarkably high, with an average Pearson coefficient of 0.95 within the diagnosis and average Pearson coefficient of 0.89 in cross-diagnosis pairs. There was a small decrease in correlation coefficient in cross-diagnosis pairs, reflecting the relatively small number of transcripts differing between cases and controls (Fig. S4), as well as the methodologic consistency of the iHLC technology to achieve uniform iHLC cultures from multiple individuals and across both diagnoses and different batches.

The transcriptome-wide DEGs significantly implicated several Wikipathways (Fig. 3A, B). The top downregulated pathways in AC iHLCs were electron transport chain (STAT 7.14), oxidative phosphorylation (STAT 5.39), purine and pyrimidine metabolism (STAT 3.4; 3.29), glycolysis and gluconeogenesis (STAT 3.28) and TCA cycle and deficiency of pyruvate dehydrogenase complex (STAT 3.1). Pathways upregulated in AC iHLC's were PI3K-Akt signaling (STAT 4.82), adipogenesis (STAT 3.8), MAPK signaling (STAT 2.8), DNA damage response (STAT 2.52) and apoptosis (STAT 2.5). These down and up-regulated pathways were statistically significant, with the $p$ values stated in Fig. 3A, B. Analysis of the DEGs against KEGG pathways, using the COLOR feature in KEGG analytics, implicated similar gene networks. For example, KEGG adipogenesis and multiple genes within that pathway were upregulated (Fig. 3C). KEGG "oxidative phosphorylation" and "electron transport chain" were again downregulated, as were multiple genes within these pathways. (Fig. 3D, E), suggesting lower ATP production.

### Decreased spare capacity of mitochondria in AC iHLCs leads to lower ATP-dependent respiration

Observations of transcriptomic signatures of downregulated oxidative phosphorylation and ATP production led us to predict that there would be functional deficits of mitochondrial bioenergetics in AC iHLCs. Both H and AC iHLCs generated ATP largely by mitochondrial respiration ($50.9 ± 6.7$pmol/min) compared to a lesser contribution from glycolysis ($9.2 ± 1.2$pmol/min) (Fig. 4A). Also, via an oxygen flux bioanalyzer, serial addition of specific mitochondrial inhibitors to block ATP production (Oligomycin), to uncouple mitochondrial respiration (FCCP) and to inhibit the electron transport chain (Rotenone + Antimycin A)

enabled quantitative measurement of specific aspects of mitochondrial bioenergetics. AC iHLCs had lower basal and maximal respiration (Fig. 4B). Basal respiration, maximal respiration, spare respiratory capacity and ATP production were significantly decreased in AC iHLCs, but AC iHLCs did not differ from H iHLCs in non-mitochondrial oxygen consumption or proton leak (Fig. 4C). If spare capacity was represented as a percentage, AC iHLCs showed significantly less spare respiratory capacity, whereas there was no difference in coupling efficiency by this approach (Fig. 4D). AC iHLCs did not differ in levels of key mitochondrial structural proteins (Complex I, II, III, V), indicating that differences in mitochondrial copy number were unlikely to explain the differences in the oxygen consumption rate (OCR) (Fig. 4E). Thus, AC iHLCs appear to be intrinsically impaired in their OCR, including regarding oxygen consumption directed to synthesis of ATP(Fig. 4F).

### Impaired metabolic flexibility in AC iHLCs

Comprehensive bioenergetic analyses of iHLCs for the substrates glutamine (GLN) and fatty acids (FAs) revealed differences between H and AC iHLCs in their dependency and capacity to oxidize these fuels (Fig. 5A). By sequentially inhibiting the specific pathway in iHLCs followed by measurement of OCR via alternative pathways, we were able to calculate the dependency of the cells on a particular pathway to meet energy demands. We found that AC iHLCs were 26.2% dependent on FAs whereas H iHLCs were 37.1% dependent (Fig. 5A). A similar trend was also observed for GLN, where AC iHLCs were 7.3% dependent and H iHLCs were 23% dependent on this fuel source (Fig. 5A). Dependency indicates that mitochondria are unable to compensate for a blocked pathway by oxidizing other fuels, whereas capacity is OCR after the two alternative pathways are blocked. For GLN, the mean mitochondrial capacity of H iHLCs was $37.73 ± 2.4$ compared to $21.17 ± 4.9$ for AC iHLCs ($P = 0.0158$) (Fig. 5A). Similarly, for FAs, mitochondrial OCR capacity was $63.41 ± 4.4$ for H iHLCs but $32.05 ± 5.39$ for AC IHLCs ($P = 0.0011$) (Fig. 5A). Based on these observations, iHLC mitochondria are highly reliant, or dependent, on FAs and GLN, and utilization of both fuels was impaired in AC iHLCs (Fig. 5B).

Next, we probed the mitochondrial function of iHLCs in the context of FA and GLN substrates. In these experiments, etomoxir ($4$ μM) was used to inhibit the synthesis of long-chain FAs (LCFAs) through inhibition of carnitine palmitoyl transferase 1a (CPT1a), and BPTES ($3$ μM) was used for inhibition of glutamine synthesis via glutaminase 1 (GLS-1) in iHLCs. These experiments provided information about basal respiration and the impact of pathway inhibition under conditions of basal substrate demand in H and AC iHLCs. These manipulations also measured the impact of pathway inhibitions on maximal respiration, reflecting the sensitivity of iHLCs to impairment of a specific metabolic pathway under conditions of high energy demand; measuring reliance on FAs or GLN to meet that demand. In the FA oxidation stress test, the source of LCFA substrates is endogenous stores of lipids and LCFAs and etomoxir ($4$ μM) is used to probe LCFA oxidation. We observed significantly decreased respiration rates of AC iHLCs (H $80.41 ± 2.5$ vs AC $50.96 ± 4.59$; $P = 0.0002$) in response to etomoxir (Fig. 5C). The spare capacity of AC IHLCs in the FA oxidation stress test was lower ($19.53 ± 3.59$) compared to H iHLCs ($73.21 ± 11.58$; $P = 0.0013$) and ATP production was also slower in AC iHLCs ($27.34 ± 5.61$) compared to H iHLCs ($51.52 ± 2.56$; $P = 0.0029$)

### Table 1 | Clinical characteristics of healthy individuals and patients with AC

| | Healthy (H) | Alcohol-associated cirrhosis (AC) |
|---|---|---|
| AGE | 51 ± 11.5 | 50.8 ± 11.07 |
| SEX | 83.3% M | 83.3% M |
| RACE | 3W-2B-1U | 5W-1B |
| BMI | NA | 29.87 ± 6.24 |
| ALT | Normal range | 24.33 ± 19.4 |
| AST | Normal range | 69.33 ± 45.65 |
| BILIRUBIN | Normal range | 12.48 ± 19.4 |
| ALKALINE PHOSPHATASE | Normal range | 131.66 ± 36.49 |
| ALBUMIN | Normal range | 3.03 ± 0.63 |
| CREATININE | Normal range | 1.85 ± 2.2 |

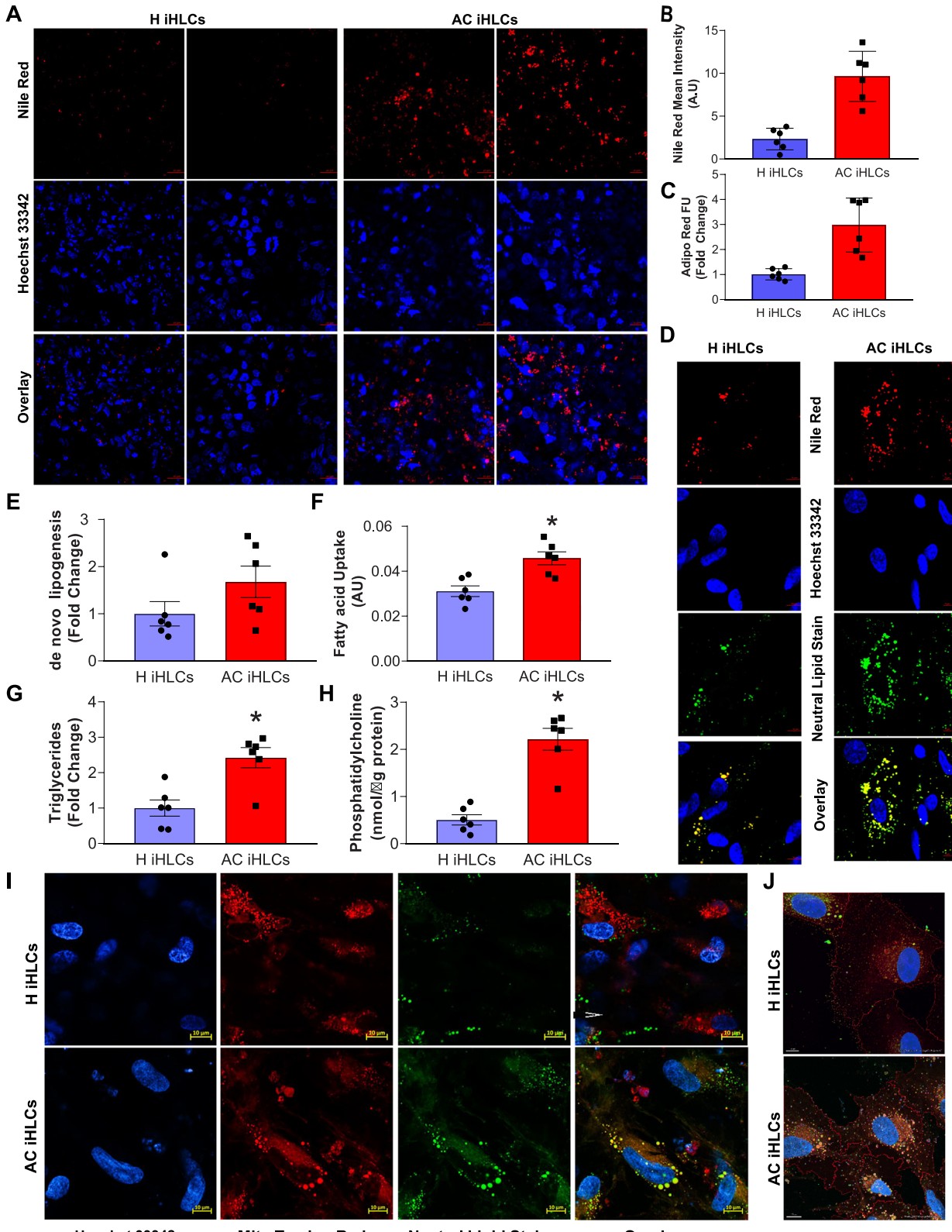

**Fig. 2 | The number of LDs in AC iHLCs is increased and the LDs are closely associated with mitochondria. A** Representative confocal images showing Nile red staining in AC and H iHLCs. Scale bar 20 μm. **B** Increased Nile red staining observed in AC iHLCs was quantified ($n$ = 6/group, *$P$ < 0.05, unpaired two tailed t-test). **C** Confirmation of increased lipid accumulation in AC iHLCs via quantitative Adipo Red assay. $n$ = 6/groups, *$P$ < 0.05(unpaired two tailed t-test). **D.** Representative confocal images for both Nile red and neutral lipid staining along with Hoechst33342 in live cells. Scale bar 5 μm Metabolic comparisons of H and AC iHLCs showed a trend for increased de novo lipogenesis (**E**) and significant increases in fatty acid uptake (**F**) and triglyceride (**G**) and phosphocholine (**H**) content. ($n$ = 6/group, *$P$ < 0.05, unpaired two tailed t-test). **I** Confocal images of mitochondrial/lipid proximity in live cells with a neutral lipid stain and the mitochondrial stain Hoechst33342. Scale bar 10 μm. **J** AIVIA machine learning generated diagram showing increased lipid droplets in proximity to mitochondria in AC iHLCs. All data were shown as mean ± SEM. All experiments were replicated three times.

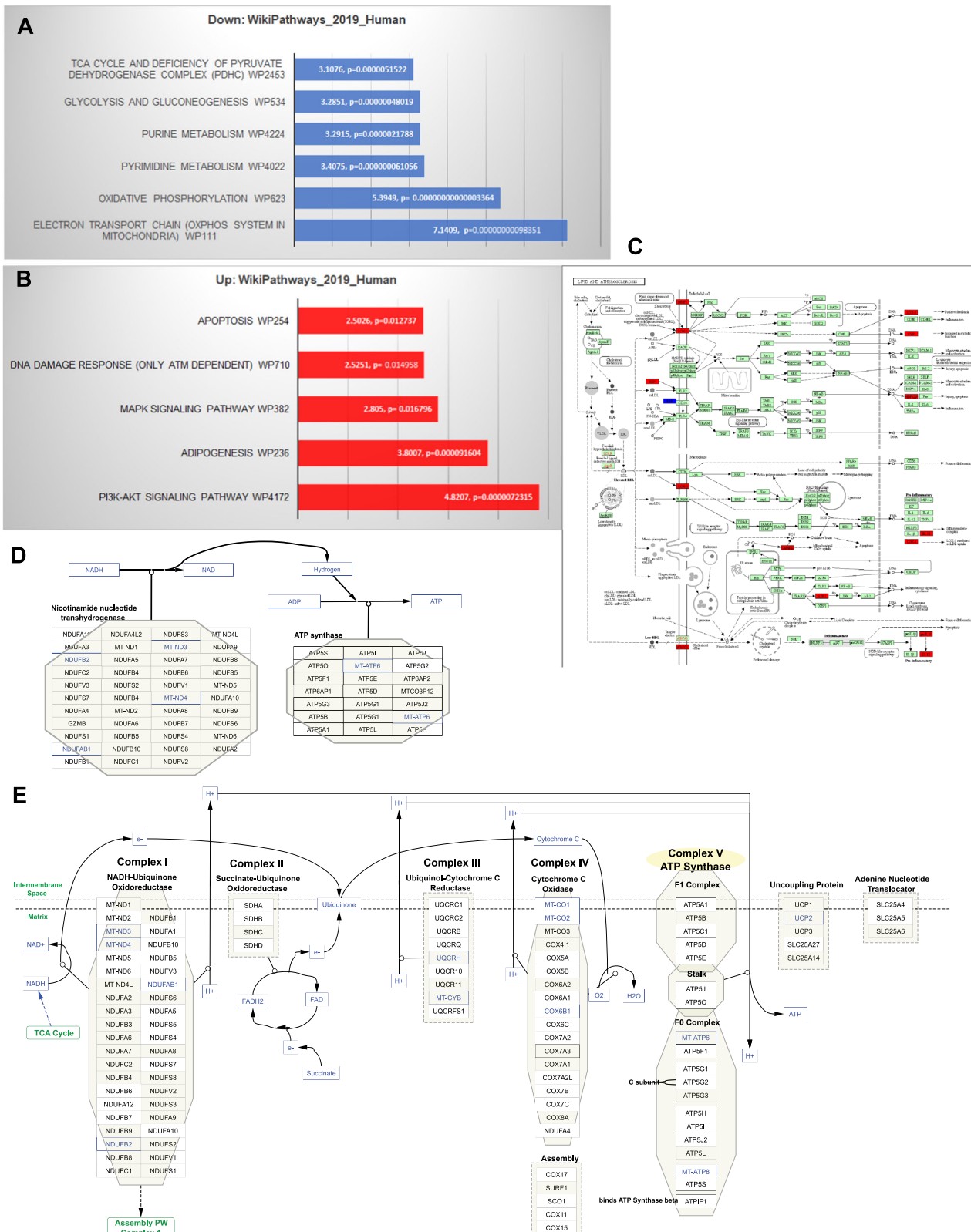

**Fig. 3 | Transcriptomic analyses of differentially expressed genes in H and AC iHLCs. A** Integrative analysis of differentially expressed genes for the Wikipathways gene set was performed using PIANO (Bioconductor). The most statistically significant downregulated pathways (blue) associated with metabolism and cell fate. **B** Wikipathways gene set analyses of upregulated pathways (red) associated with metabolism and cell fate. **C** Analyses of differentially expressed genes for KEGG pathways using the COLOR feature. Several genes associated with lipid pathways (red) were upregulated. **D** KEGG COLOR analyses highlighted multiple genes interacting with oxidative phosphorylation that were downregulated(blue), predicting lower NAD and ATP (blue). **E** KEGG COLOR analyses also highlighted the electron transport chain in which several genes interacting with mitochondrial complexes were downregulated (blue) in AC.

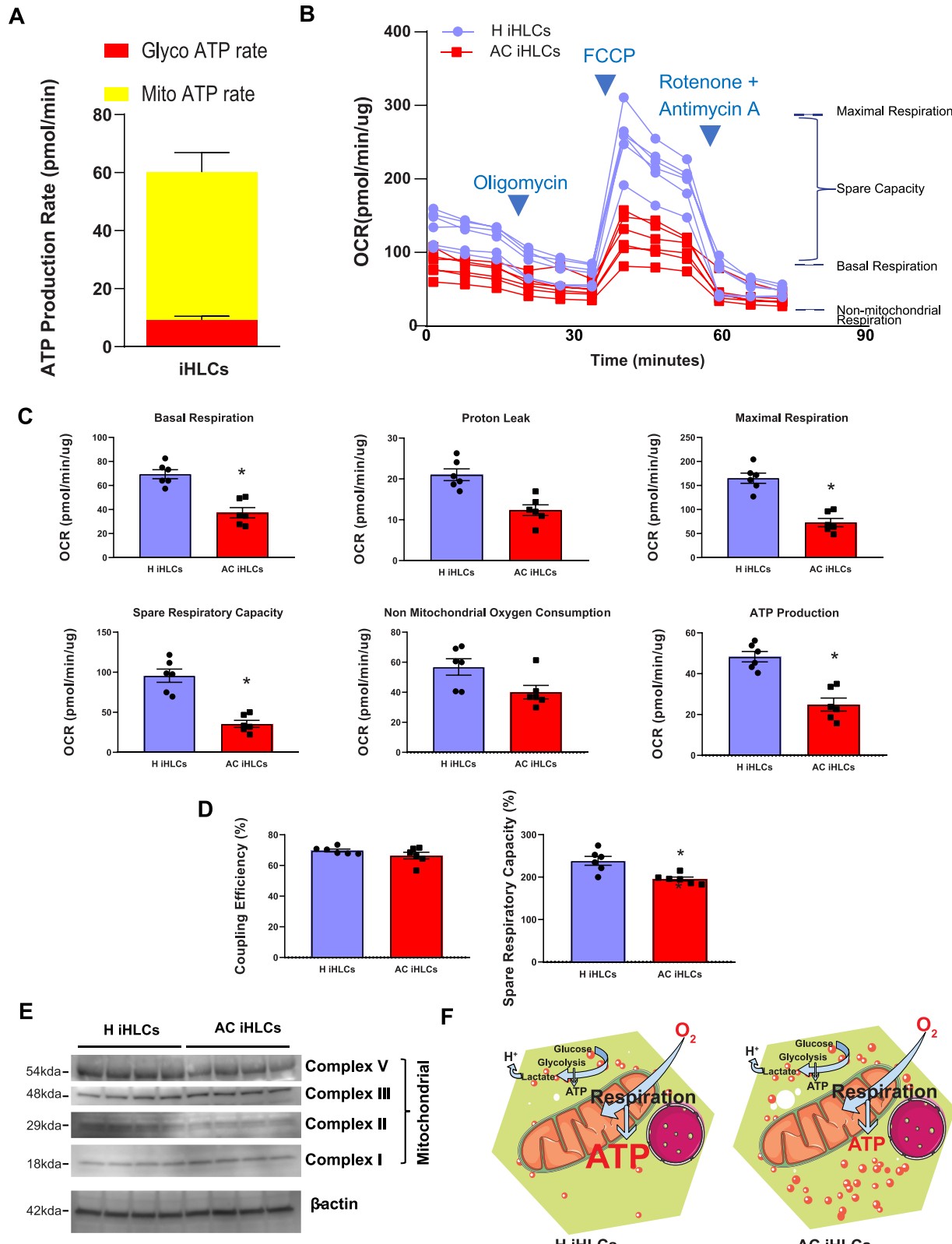

(Fig. 5C, D). Comparing the maximum respiration rates of H and AC iHLC's revealed that with FAs as substrates H iHLCs had a higher maximum respiration rate (153.6 ± 11.64) as compared to AC iHLCs (70.49 ± 7.05, $P = 0.0001$), indicating again that AC iHLCs had a reduced ability to metabolize intracellular FA (Fig. 5D, E).

Similarly, glutamine oxidation was evaluated via stress test under a saturating concentration of glutamine (2 mM) and then using BPTES

for inhibition (Fig. 5F). We observed a significant ($P = 0.0055$) lower basal respiration rate (30.3 ± 2.12) in AC iHLCs compared to H iHLCs (46.89 ± 4.2) (Fig. 5G). Furthermore, the spare capacity of GLN oxidation in AC iHLCs was lower (28.0 ± 2.06) compared to H iHLCs (61.33 ± 4.6; $P < 0.0001$), and ATP production rate was also slower in AC iHLCs (16.76 ± 1.59) compared to H iHLCs (29.1 ± 2.9; $P = 0.0039$). In response to high energy demand, AC iHLCs showed a lower ability to

**Fig. 4 | Bioenergetics mitochondrial is significantly altered iHLCs. A** Relative anaerobic (Glycolytic) and aerobic (Mitochondrial) ATP production by iHLCs. **B** Comparison of H and AC iHLC mitochondrial bioenergetics using the Seahorse instrument demonstrating lower mitochondrial efficiency. **C** Quantification of mitochondrial functional parameters showed that basal respiration, maximal respiration, spare respiratory capacity, and ATP production were significantly decreased in AC iHLCs compared to H iHLCs. AC iHLCs had lower spare mitochondrial capacity (62%; $P < 0.0001$) as well as lower basal (46%; $P = 0.0002$), maximal (55%; $P < 0.0001$), and ATP production (48%; $P = 0.0002$)(unpaired two-tailed t-test). There was no statistical difference in proton leak and non-mitochondrial oxygen consumption (unpaired two tailed $t$ test) **D** AC iHLCs were also significantly lower in percentage spare respiratory capacity (18%, $P = 0.0041$) but did not differ in coupling efficiency (unpaired two-tailed t-test). **E** Immunoblot analyses of mitochondrial complex I, II, III and V normalized with β-actin in cell lysates from samples measured for OCR. **F** Schematic diagram depicting the differential mitochondrial bioenergetics observed in H and AC iHLCs. All data were shown as mean ± SEM ($n = 6$/group).

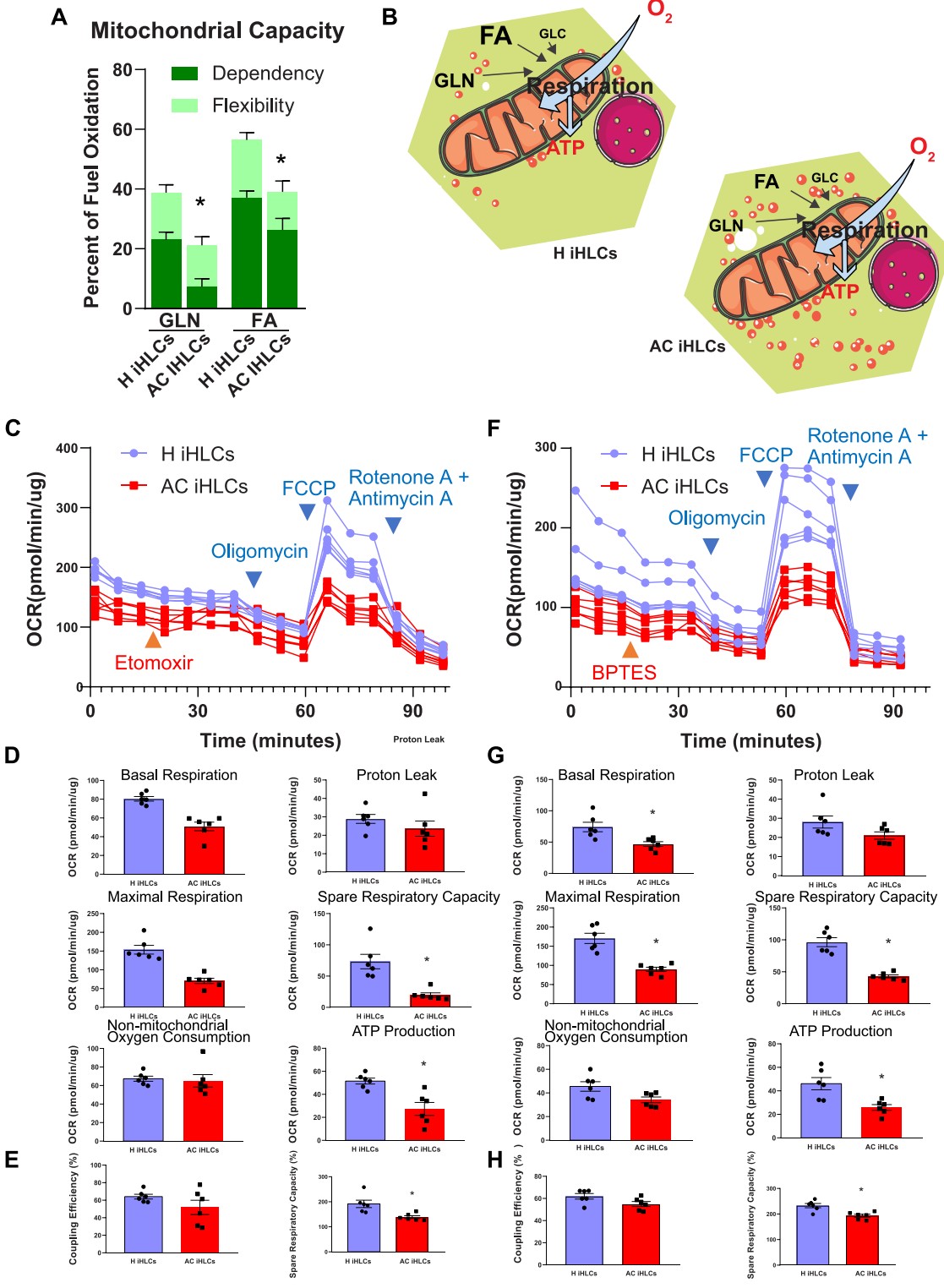

**Fig. 5 | Utilization of fatty acid and glutamine substrates by mitochondria is diminished in AC iHLCs leading to lower mitochondrial capacity. A** Dependency of mitochondrial metabolism on glutamine (GLN) (68%, $P = 0.0018$) and fatty acid (FA) (29%, $P = 0.035$) substrates was reduced in AC iHLCs. The changes in flexibility were not statistically significant. However, the mitochondrial capacities, a measure for both, differed significantly (unpaired two tailed $t$ test). **B** Schematic diagram of substrate utilization by H and AC iHLCs. **C** Substrate oxidation stress test (OCR) profiles of respiration parameters critical for substrate demand in H and AC iHLCs. For OCR, compounds were sequentially injected measuring basal respiration, acute response to the inhibitor etomoxir (blocking fatty acid transport to mitochondria), and maximal respiration. **D** Quantitative measurements of parameters demonstrating decreases in basal respiration (37%, $P = 0.0002$), maximal respiration (54%, $P = 0.0001$), spare respiratory capacity (73%, $P = 0.0013$), and ATP production (47%, $P = 0.0029$) in AC iHLCs compared to H iHLCs after FA substrate blockade whereas there were no differences between case and control iHLCs in proton leak and nonmitochondrial oxygen consumption. (unpaired two tailed $t$-test). **E** Coupling efficiency and spare respiratory capacity expressed as percent for AC IHLCs for FA substrate (unpaired two tailed t-test). **F** Substrate oxidation stress test OCR profiles critical for substrate demand in control and AC iHLCs showing basal respiration and acute response to BTES (which blocks GLN transport to mitochondria) and maximal respiration. **G** Quantitative measurements of parameters demonstrating decreases in basal respiration (35%, $P = 0.0055$), maximal respiration (46%, $P = 0.0002$), spare respiratory capacity (68%, $P < 0.0001$), and ATP production (42%, $P = 0.0039$) in AC iHLCs compared to control iHLCs after GLN substrate blockade whereas no differences were observed between case and control iHLCs in proton leak and non-mitochondrial oxygen consumption. (unpaired two-tailed t-test). **H** Quantification of Coupling efficiency and spare respiratory capacity is expressed as percent for AC iHLCs for GLN substrate. All data were shown as mean ± SEM ($n = 6$/group).

use glutamine as the maximum respiration rate of AC was 58.27 ± 4.2 compared to 108.2 ± 7.86 for H iHLCs ($P = 0.0002$) (Fig. 5G, H).

It is important to point out that spare capacities measured as percentages in both FA and GLN oxidation also differed between case (AC) and control (H) iHLCs (FA: H 191.6 ± 14.74 AC 138.7 ± 5.66, $P = 0.0073$; GLN: H 232.9 ± 8.34 AC 193.4 ± 5.81, $P = 0.0030$), whereas coupling efficiencies did not (Fig. 5E, H). A bioenergetic health index (BHI), was proposed in 2014 and involved the generation of a formula using the four parameters of reserve capacity, ATP production, non-mitochondrial OCR and protein leak OCR[14]. Our analyses did not demonstrate any difference in BHI between H iHLCs and AC iHLCs (Fig. S5).

### Aramchol reduces LD accumulation in AC iHLCs and improves senescence

Aramchol, a partial inhibitor of hepatic stearoyl-CoA desaturase (SCD), reduced steatohepatitis and fibrosis in animal models and reduced steatosis in an early clinical trial of NASH[15]. Our transcriptome data indicated that there was an increase in average SCD gene transcripts in AC iHLCs compared to H iHLCs Aramchol. We found that after treatment with Aramchol (10 μM for 72 h) LD content decreased in all AC iHLCs compared to vehicle treatment (Fig. 6A). There was also a nonsignificant reduction in LDs, but no marked change, when H iHLCs were treated with Aramchol (Fig. S6)

Given the increased LD content and its linkage to diminished mitochondrial function in AC iHLCs, we next tested whether Aramchol could improve mitochondrial function. Aramchol Treatment of AC iHLCS with Aramchol improved four critical parameters of mitochondrial function; namely, basal respiration, maximal respiration, spare respiratory capacity and ATP production (Fig. 6B). To understand the potential role of LDs to accelerate cellular aging of AC iHLCs, we cultured these cells for an additional four weeks and measured the senescence marker beta-galactosidase. Aramchol treatment significantly reduced the senescent population of AC iHLCs from 83.22 ± 2.1% to 56.93 ± 4% ($P = 0.0002$) (Fig. 6C).

### Role of *PNPLA3* and *MARC1* in mitochondrial-related oxidative damage and LD size content

Investigating *PNPLA3* and *MARC1*, two genes implicated by genomic studies of AC, we found evidence for dysregulated subcellular localization and dynamics of both protein encoded by these genes in AC iHLCs. *PNPLA3*, which encodes adiponutrin, is a gene of relatively strong effect in AC as shown by GWAS in multiple cohorts[12]. *MARC1*, a mitochondria-localized protein, and its gene has also been implicated AC risk gene[13]. In AC iHLCs PNPLA3 and MARC1 co-localized with each other, and both were closely associated with mitochondria (Fig. 7A) and LDs (Fig. 7B). MARC1 is well known for its localization in the outer membrane of mitochondria[16]. Colocalization of PNPLA3 and LDs with MARC1 and mitochondria was less pronounced in H iHLCs. Similarly, the colocalization of PNPLA3 and MARC1 was enhanced in liver tissues

of patients with AC (Fig. 7C), leading to a model in which recruitment of PNPLA3 to LDs and MARC1 to/mitochondria may play a role in the pathogenesis of AC (Fig. 7D).

To understand the physiological function of PNPLA3 in iHLCs, we knocked down *PNPLA3* expression via siRNA. *PNPLA3* inhbition lead to significantly larger LD size in AC iHLCs compared to scrambled AC treated with scrambled siRNA (Fig. 8A, B). By applying intelligence-machine learning to multiple levels of 3D confocal images, we found a significant increase in LDs size and volume (large and medium LD) after *PNPLA3* knockdown (Fig. 8B, C).

Transcriptomic analysis of AC iHLCs suggested that at baseline these cells experience more oxidative damage (Fig S7), representing a potential mechanism for impairment of mitochondrial OCR. To test this hypothesis, we quantified lipid peroxidation in live cells via flow cytometry. Congruent with the functional impairments that we observed in AC mitochondria, AC iHLCs had 2.4-fold (69,325 ± 9732 vs 167,826 ± 3653, $P < 0.0001$) higher levels of lipid peroxidation (Fig. 8D). Lipid peroxidation tends to correlate with protein adducts generated by products of lipid oxidation and that can partially or completely disable proteins. Therefore, we screened eight fatty acid metabolism enzymes for oxidative damage, two of these enzymes are involved in the β-oxidation of fatty acids, ACSS2, a cytosolic enzyme that activates acetate for use in lipid synthesis, and ACSL1, converts LCFAs to acyl CoA. Both were significantly modified by HNE adducts in AC iHLCs (Fig. 8E). These protein modifications further suggest that oxidative damage may play a role in altered mitochondrial bioenergetics and cellular senescence in AC.

Next, we explored the source of ROS as mitochondria are a major source of ROS generation, especially when they are dysfunctional. We measured the antioxidant capacity from the same set of mitochondria and observed reduced glutathione reserve in AC iHLCs compared to H iHLCs (Fig. S8A). These data suggest, that AC iHLCs may generate more ROS and have a reduced antioxidant capacity compared to H iHLCs, leading to greater oxidative stress in the former cells. We further investigated the possible role of supercomplex assembly in reduced ATP production. Preliminary analyses of native PAGE of mitochondrial membrane fraction showed a mobility shift in super complex (Fig. 8B). However, a comprehensive future studies is required to understand the assembly process in details. Notably, there was no significant difference in mtDNA content between H iHLCs and AC iHLCs (Fig. S9).

## Discussion
Exogenous alcohol is primarily metabolized by parenchymal hepatocytes that constitute ~80% of liver mass[17]. Furthermore, numerous studies, including ours, in both humans and complementary animal models, have demonstrated that hepatocytes drive increased fatty acid metabolism and accumulate lipids in response to heavy alcohol consumption[18,19]. Other liver cell types, including stellate cells, Kupffer cells and cholangiocytes, also play critical mechanistic roles in ALD[20,21]. Although fatty liver has been powerfully implicated as a

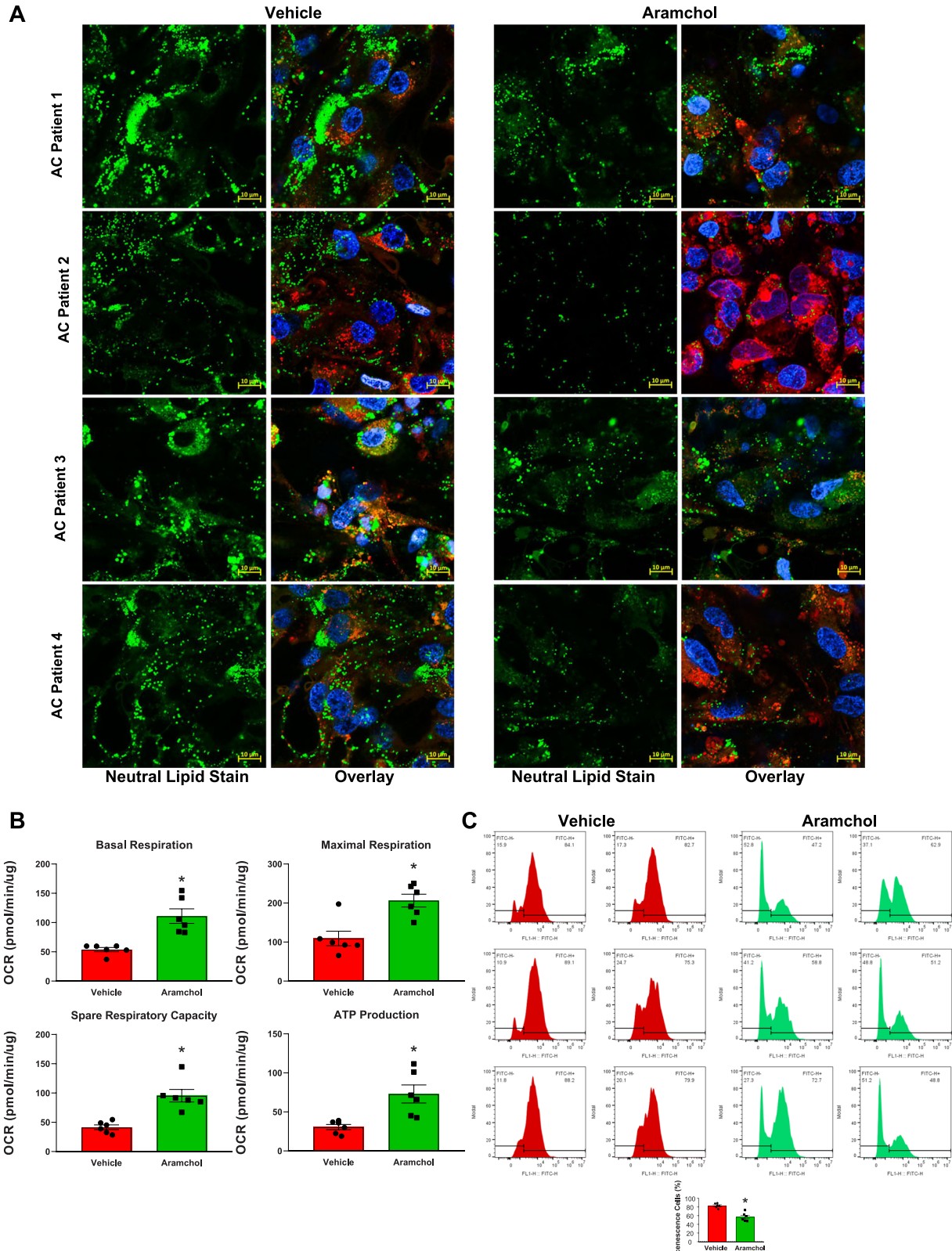

**Fig. 6 | Treatment of AC iHLCs with Aramchol leads to decreased LD formation, increased mitochondrial performance and diminished senescence.**
**A** Representative confocal image (40× objective) of AC iHLCs treated with vehicle or Aramchol for Neutral lipid stain (green), and nuclear stain (blue). Scale bar 10 μm. **B** Quantitative measurements of parameters demonstrating increases in basal respiration (106%, $P = 0.0013$), maximal respiration (88.9%, $P = 0.0028$), spare

respiratory capacity (130%, $P = 0.0008$), and ATP production (139%, $P = 0.0055$) in AC iHLCs compared to control iHLCs (unpaired two tailed t-test). **C** Flow cytometry histogram of senescence cell staining of six AC patient-derived iHLCs(red) and after treatment of the same group of samples with Aramchol (green). Senescence-positive cells decreased 32% ($P = 0.002$) after treatment with Aramchol (unpaired two-tailed $t$ test). All data were shown as mean ± SEM ($n = 6$/group).

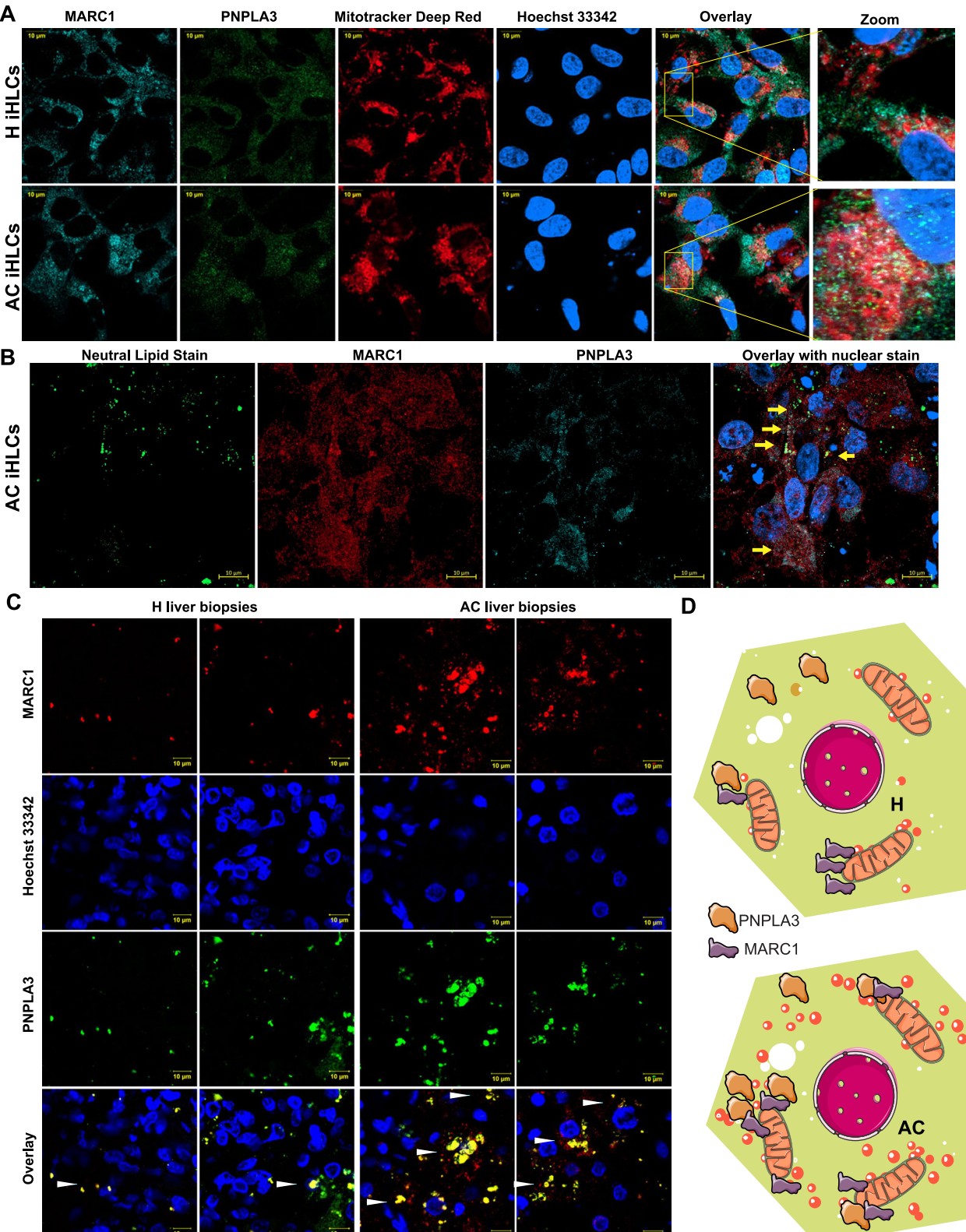

**Fig. 7 | In AC iHLCs, PNPLA3 and MARC1 show increased colocalization with mitochondria and LDs. A** Representative confocal image (60× objective) of control and AC iHLCs for MARC1 (cyan), PNPLA3 (green), mitochondria (red) and nuclear stain (blue). Scale bar 10 μm. **B** Representative confocal images (40× objective) of control and AC iHLCs for neutral lipid stain (green), MARC1 (red), PNPLA3 (turquoise), and nuclear stain (blue). Scale bar 10 μm **C** Increased colocalization of MARC1 and PNPLA3 in AC iHLCs compared to control iHLCs. Scale bar 10 μm. **D** Schematic model of interaction of PNPLA3 with MARC1, and in the physical context of proximity of mitochondria and lipid droplets. All data were shown as mean ± SEM($n = 6$/group).

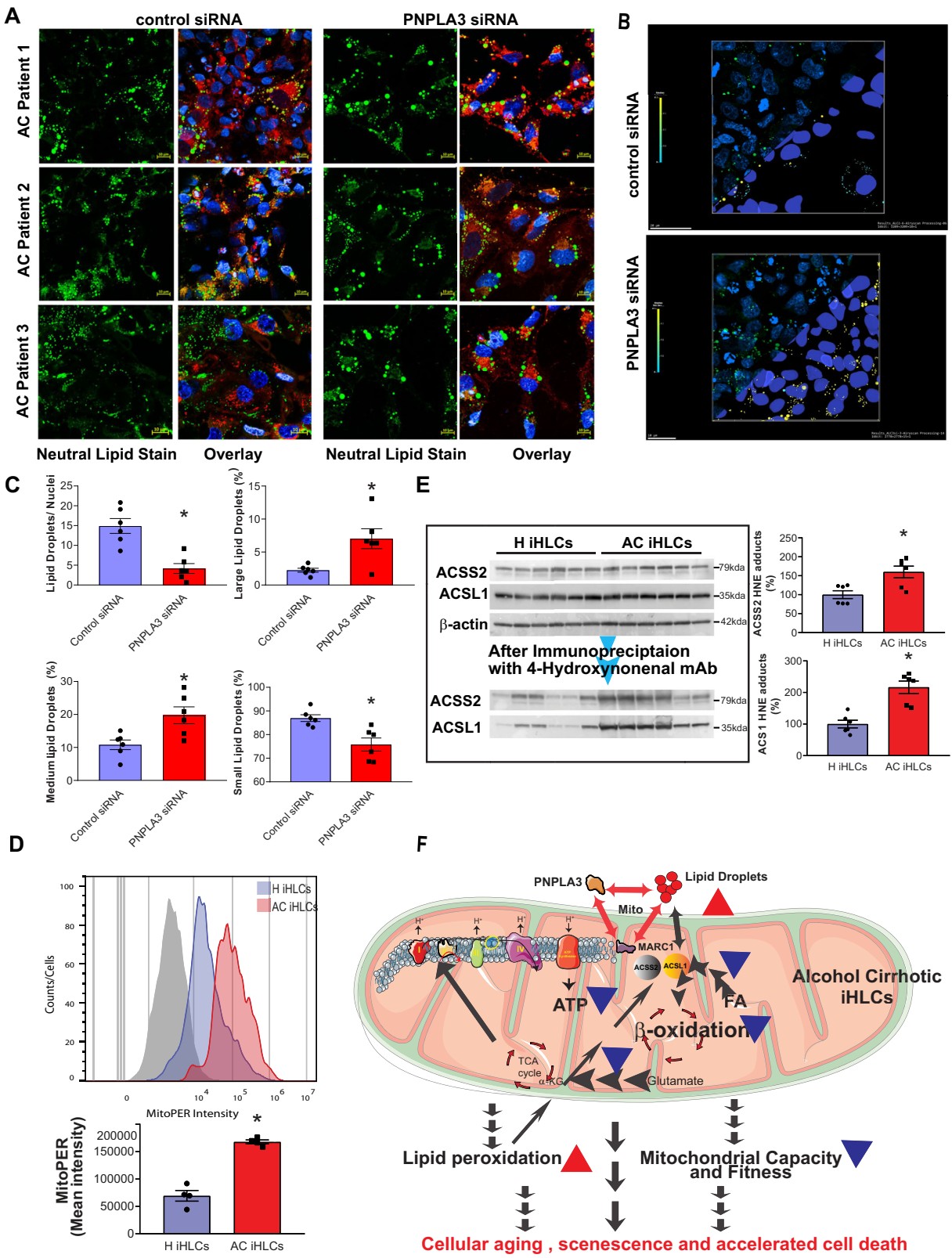

critical step in the progression to ALD/AC, the underlying mechanisms of fat accumulation, or indeed whether patients with ALD/AC are more prone to accumulation of LDs, is unknown. However, among the top genes identified in AC-focused GWASs are three genes that are plausibly involved in intrahepatocyte LD formation, including *PNPLA3*, which is the leading gene emergent from such genetic

studies. These findings suggest that LD accumulation is not only an early, critical event in the pathogenesis of AC and ALD but that people suffering from ALD are more likely to be innately predisposed to LD accumulation, in addition to whatever genetic differences they carry that might affect the function of other cell types and organs that contribute to AC pathogenesis.

**Fig. 8 | The interplay of PNPLA3 and MARC1 with mitochondria and LDs in the interactome lead to increased lipid peroxidation and oxidative modification of lipid metabolism proteins in AC iHLCs. A** Representative confocal images of iHLCs derived from three patients with AC, the cells having been stained with neutral lipid stain (green) and nuclear (blue). AC IHLCs were treated with either scrambled siRNA or PNPLA3 smart pool siRNA and LD were visualized by airyscan confocal technology. Scale bar 10 μm. **B** Representative machine learning clipping plane images from control and PNPLA3 siRNA treated AC iHLCs. **C** Quantification of LD size/nuclei, and volume of LD based on sizing criteria: small, medium and large, as described in methods. Knockdown of *PNPLA3* with siRNA led to increased LD size ($P = 0.007, 0.0111, 0.0121$ and $0.0050$; unpaired two-tailed $t$-test). **D** Flow cytometric quantification of Mitochondrial lipid peroxidation using MitoPER. Representative univariate histograms of H and AC iHLCs along with no-reagent control (gray shade) in the top panel. The quantitative measurement of the mitochondrial lipid peroxidation in the bottom panels demonstrated a significant increase in AC iHLCs, ($n = 4$/group, unpaired two-tailed $t$-test, $P < 0.0001$). **E** Selective modification of short and long-chain derived fatty acyl-COA synthetase by HNE adducts represented by western blot analyses in the left panel. Quantitative measurement of HNE adducts with ACSS2 and ACSL1 showed a significant increase in AC iHLCs compared to H iHLCs ($P = 0.0089$ and $0.0005$; unpaired two tailed $t$-test). **F** Schematic diagram of a hypothesized underlying mechanism for lower AC iHLCs bioenergetic fitness due to the interplay of PNPLA3 and MARC1 with mitochondria and lipid droplets generating lipid peroxidation and mitochondrial dysfunction. Oxidative modification of mitochondrial proteins and key metabolic proteins can lead to reduced mitochondrial metabolism of fatty acids and glutamate leading to mitochondrial-reduced capacity and efficiency and decreased ATP production and overall. to the poor bioenergetic state of AC iHLCs. All data were shown as mean ± SEM ($n = 6$/groups except stated).

One of the major observations in our study was the beneficial effect of Aramchol, a partial inhibitor of hepatic stearoyl-CoA desaturase, on LD formation in AC iHLCs and on cellular aging Recently, Aramchol showed positive effects in a clinical trial for NASH[15]. Aramchol has not yet been tested in ASH/ALD models, but given the results here demonstrating a positive effect on LD accumulation and on impaired bioenergetics in AC iHLCs there is now an more incentive to evaluate AAramchol for ALD. Also, our in vitro preclinical cellular model could be utilized as a platform to test other drugs for efficacy in ALD. for ALD.

In that regard it is important to note that nonparenchymal cells, including Kupffer and stellate cells, play a major role in inflammation and fibrosis in AC. And it is known that Aramchol also modulates fibrogenesis via stellate cells[22]. In future studies we can explore the effect of Aramchol on the inflammatory milieu during the progression of AC, or use multi-cellular organoids to study the role of other cells in AC progression and what effect Aramchol might have on cellular crosstalk with hepatocytes.

In addition to genetic variations that can determine vulnerability to consequences of heavy drinking, and that modulate risk of heavy drinking, simultaneous intake of a high caloric diet in rodents and nonhuman primates has been shown to act as a second "hit" to induce ALD. These observations led us to hypothesize that AC iHLCs would show intrinsically disturbed metabolic and transcriptomic features compared to H iHLCs. Although such differences might lead to ALD only in the presence of other liver cell types and extrahepatic factors, or in the context of hepatic cytoarchitecture, iHLCs offer a simplified cellular model to capture innate differences and consequences intrinsic to hepatocytes. Therefore, while several aspects of ALD cannot be captured by studies of isolated iHLCs, multiple parameters of cell morphology and its omics, including transcriptomics, proteomics and metabolomics, are accessible. Indeed, we have observed that readouts in each of these domains validated the utility of iHLCs as a model of intrinsic hepatocyte function, as well as hepatocyte dysfunction that was present in AC.

A major observation of this study is that iHLCs not only take up FAs, leading to FA accumulation in LDs, but AC iHLCs accumulate dramatically more LDs compared to HiHLCs. This striking difference can be attributable to multiple causes; however, the increased lipid accumulation correlating with defects in OCR and mitochondrial function, suggest that a deficiency in fat metabolism may contribute to the progression of AC. In AC iHLCs this difference is most likely due to a genetic predisposition, although this notion is still unproven, as it would require genetic manipulation of the cells, for example by CRISPR, in a gain-of-function approach to determine if the presence of a given AC-related SNP or mutation is sufficient to convert otherwise wild-type iHLCs toward an AC phenotype. Here, we have approached that issue via knockdown of *PNPLA3* expression, which did exacerbate LD formation. However, genotype-based prediction or stratification (so-called Mendelian randomization) may be premature for AC and

ALD. Via GWAS we and others have shown that polygenic background is critical to risk of ALD[12]. However, gene scores or polygenic scores derived from these studies are as yet modestly predictive, as compared to the functional ddifferences derived between AC iHLCs and H iHLCs observed in this study Notably, several AC-associated genes that have been well-replicated; namely, *PNPLA3, HSD17B13, FAF2, SERPINA1* and *SUGP1*, are expected to alter LD accumulation, but the odds ratios for each of these genes in AC small (ORs: 2.19, 0.57, 0.61, 1.9 and 1.49, respectively)[12].

The co-localization of LDs with mitochondria has been widely observed in adipocytes[23] but has also been reported in hepatocytes in the context of liver diseases, including both NAFLD and ALD[24]. In the ACSL1 hepatic cell line, peri-mitochondrial LDs appear to play a dynamic role[25]; however, little is known about their function and specific roles in lipid metabolism and bioenergetics in hepatocytes. It is important to note that electron-dense structures of mitochondria-associated LDs from mouse hepatocytes are morphologically distinct from the structures found in LD-anchored mitochondria (LDAM) or peridroplet mitochondria (PDM) reported in other cell types, including adipocytes[24]. Here, we find a greater number of mitochondria-associated LDs in AC iHLCs compared to H iHLCs cells, suggesting that they may have a role in AC progression. If so, iHLCs would offer a powerful cellular model to study, and potentially manipulate or block, the formation of the mitochondria-associated LDs.

Transcriptomic comparison of AC and H iHLCs revealed 343 statistically significant DEGs. The overall high consistency of transcript abundance between iHLCs from different batches and between iHLCs from cases and controls points to the consistency of the iHLC model such that pathways implicated by DEGs between AC and HiHLCs are likely to be specific to disease conditions, or other random biological variations that happened to stratify between cases and controls, rather than technical confounds. Furthermore, the dysregulated pathways are logically tied to AC, as would not be expected of random biological variation, and in addition, these pathways were directionally upregulated or downregulated in a biologically coherent way. The top gene networks implicated in AC iHLCs were upregulated lipid synthesis and adipogenesis, apoptosis and signaling pathways for cell death, while oxidative phosphorylation and electron transport chain were downregulated. Upregulation of the lipid pathway and cell death pathway transcripts converges with the increased cellular uptake of FAs, a trend for increased de novo lipogenesis and increased content of triglycerides and phospholipids that we observed in AC iHLCs, and with their increased numbers of mitochondria-associated LDs.

The most salient aspect of the AC iHLC transcriptome was downregulation of cellular bioenergetics. The KEGG pathway finding of lower NAD and ATP levels in AC iHLCs has many interesting implications. Earlier studies indicated a direct association of reduced NAD levels with alcohol metabolism and metabolic disorder[26,27]. In recent years lower NAD has been linked to accelerated aging[28]. These findings drove our further experiments on mitochondrial respiration that we

assessed via measurements of OCR. The iHLCs primarily generated ATP via mitochondrial respiration whereas anaerobic glycolysis had a smaller contribution. Therefore, mitochondrial respiration parameters are good indicators of cellular metabolism and fitness of hepatocytes. To the best of our knowledge this is the first report of mitochondrial bioenergetics comparing primary hepatocytes or iHLCs from healthy individuals and patients with AC, but the observations we made are not unexpected, given that GWASs have pointed to inherited variation in LD-associated genes as predisposing to AC.

Mitochondrial bioenergetics in AC iHLCs are divergent in several interrelated ways, as would be expected if many of these differences are ultimately tied to mitochondrial function. The first, and probably foundational, difference was a decreased basal OCR in AC iHLCs. Because of parallel increases in lipid peroxidation and increased levels of cell death transcripts, this difference can be interpreted as indicating weaker cellular health and suggests that under some conditions AC iHLCs might not be able to sustain sufficient oxidative phosphorylation to meet cellular energy demands. Cells were then administered oligomycin, an inhibitor of ATP synthase, to determine ATP-linked OCR and proton leak. The purpose of this step was to inhibit proton flux across the inner mitochondrial membrane via electron transport through Complexes I-IV, and thus under this proton leak condition the OCR is decreased. There were no differences between AC and H in terms of proton leak, whereas ATP production linked to the OCR was significantly lower in AC iHLCs compared to H iHLCs. The uncoupler FCCP was used to measure the maximal respiration, and a high OCR at this stage compared to the basal OCR demonstrates that the mitochondria are using less than the maximum rate of electron transport under the current supply of fuel. Because the AC iHLCs had a lower maximal respiration it can be inferred that their electron transport cannot reach the same level as H iHLCs with the same supply of fuel. One of the possible hypothesis is that the difference of mitochondrial super complexes between two groups as we observed variations of mobilty in native PAGE from mitochondrial membrane super complexes. However, a comprehensive future investigation is necessary to elucidate the exact mechanism behind such mobility shift and the process of supercomplex assembly.

The difference between basal and maximal OCR is known as spare or reserve bioenergetic capacity[29]. We observed significantly less spare capacity of AC iHLCs compared to H iHLCs, as is coherent with numerous studies showing that mitochondrial spare capacity is reduced or depleted under stress conditions, including cardiovascular diseases, inflammation, aging and cancer[30–33]. Interestingly, when rotenone and antimycin A were added to measure non-mitochondrial OCR, there was no difference between H and AC iHLCs, a sign that other cellular oxidative reactions linked to energy metabolism may not be disturbed in AC, or at least that such disturbances are not captured in isolated hepatocytes as modeled by iHLCs.

The determination of metabolic functional parameters from patient-derived hepatocytes may give significant clues to disease progression. It has been proposed that a patient's composite mitochondrial profile should be determined as an index of their bioenergetic health (the so-called BHI)[14]. This proposal may not have gained momentum due to multiple technical challenges, such as difficulty in generating a homogeneous primary hepatocyte cell population from patients. However, it may also be necessary to use different or more parameters to compute BHI or to compute different type of BHI. Via OCR analyses, we did not observe any differences in BHI. Potentially, iHLCs noninvasively derived from peripheral blood could be used to detect early indicators of vulnerability with regards to bioenergetic health. It is important to note that glycolysis had a small contribution to ATP production in iHLCs. However, its role in the disease microenvironment and limited substrate environment in AC might be crucial in terms of both bioenergetics management and oxidative stress generation.

Aramchol, a partial inhibitor of hepatic stearoyl-CoA desaturase (SCD1), has improved steatohepatitis in animal models and has reduced steatosis in patients with NASH in a clinical trial[15]. As noted earlier, Aramchol also acts via stellate cells[22]. We found that AC iHLCs treated with Aramchol, not only reduced lipid droplet formation but also protected against cellular senescence. Testing in iHLCs of other drugs intended to improve hepatocyte function and to treat or prevent liver disease should be a future and might also be used in clinical settings to target treatments to patients most likely to respond. Hepatocytes produce ATP primarily through oxidative phosphorylation using fuels, such as GLN and FA, that are taken up by cells and oxidized. For example, acetyl CoA, which is generated by beta-oxidation of FA, is further oxidized in the TCA cycle to generate NADH and FADH2. Dysfunctional FA metabolism and energy generation in hepatocytes have been widely recognized in liver diseases[34,35] but can be consequent to some other pathogenic process (*e.g.*, viral cirrhosis) rather than causal. In this study, we observed a distinct bioenergetic signature of reduced mitochondrial capacity, oxidative FA flux and reductions in mitochondrial fitness parameters of isolated AC iHLCs. Use of these cells allowed us to investigate in detail particular aspects of metabolism that are not easily studied in vivo due to confounding effects of disease or treatments. Notably, we found that GLN is a key mitochondrial energy substrate of iHLCs, and both capacity and mitochondrial fitness parameters with the GLN substrate were significantly lower in AC iHLCs compared to control cells.

Genetic variations in *PNPLA3* were first implicated in liver diseases, including NAFLD and ALD, more than a decade ago[36,37], and recently large GWASs, including a study that included our NIAAA cohort, have identified variations in *PNPLA3* as having the largest effect in AC[12]. Overexpression of the *PNPLA3* risk allele in mouse liver leads to hepatic steatosis[38], whereas deletion of the gene did not cause fatty liver disease or metabolic syndrome[39]. *MARC1*, which encodes for mitochondrial amidoxime reducing component 1, is a second gene involved in LD formation that has been implicated by GWAS in ASH and NASH without the benefit of a prior hypothesis (ref. [13]). SNPs in *MARC1* are also associated with an increase in hepatic phospholipids and with the severity of NAFLD[40]. Although the function of MARC1 is not well known, it can generate nitric oxide from nitrite, causing oxidative stress[41]. In this study, we found that both PNPLA3 and MARC1 are colocalized with LDs and mitochondria in AC hepatocytes. Thus, in iHLCs the proteins encoded by the two genes with the largest effect in AC GWAS colocalize in the microenvironment of LD-associated mitochondria. *PNPLA3* knockdown in AC iHLCs led to decreased mitochondrial contact with LDs and increased LD number/nuclei. We used machine learning to address this critical aspect of role of PNPLA3. By further analyses of LD volume in those samples we found that both large- and medium-volume lipid droplets were significantly increased in number, whereas small LDs were decreased, in AC iHLCs with knockdown of *PNPLA3* compared to HiHLCs. In a mouse model the Pnpla3148M/M mutation also led to increased in both number and increased LDs[42], showing the relevance of human-derived iHLCs as a cellular model bridging human and mouse in vivo studies. Understanding the exact roles of these genes and their variants in LD formation is beyond the scope of this study, but clearly the iHLC model can be a useful tool in deciphering their pathogenic roles, and potentially in reversing them.

These data from iHLCs lend support to the idea that oxidative stress plays a pathogenic role in AC[43]. Oxidative stress has multiple roles in the pathogenesis of ALD[44,45]. One of its roles is the oxidative modification of proteins and consequent dysfunction of metabolic pathways. In mitochondria, superoxide is generated by electrons leaking from the electron transport chain complexes I and III that are captured by molecular oxygen[46]. In addition, we observed a lower antioxidant capacity of the mitochondria, particularly in reduced glutathione, in AC iHLCs compared to H iHLCs. Recently mitochondrial

glutathione was shown to have a major role in disease pathogenesis[47]. A dysfunctional downstream electron transport chain leads to increased production of superoxide[48]. The per-hydroxyl and hydroxyl radicals from hydrogen peroxide/superoxide are highly reactive, rapidly peroxidizing lipids[49,50], thereby leading to oxidative damage of other types of molecules. By a direct readout of mitochondrial oxidative stress in live-cell flow cytometry analyses of mitochondrial lipid peroxidation, we found that there is increased oxidative stress levels in AC-derived hepatocytes. Beyond observing a transcriptomic signature of oxidative stress, we went further to test the footprint of oxidative stress by detecting lipid oxidation, such as HNE adduct formation, in the protein molecules. Importantly, two proteins, ACSS2 and ASCL1, that are involved both in FA metabolism and mitochondrial bioenergetics, showed greater oxidative damage in AC iHLCs compared to H iHLCs. Although the pathogenic role of these proteins and the oxidative adducts l need to be further, these observations suggest that oxidative stress is a pathway that might lead to LD accumulation in AC iHLCs.

Use of patient-derived iPSC-induced AC iHLCs represents cellular model preserving differences in the transcriptomic, bioenergetic and morphological landscape of AC hepatocytes. Transcriptomic and other functional analyses of AC iHLCs revealed impaired oxidative phosphorylation and ATP production in AC (Fig. 8F). Furthermore, the bioenergetics of AC iHLCs are biased towards FA biosynthesis and accumulation of LDs, which are in turn mainly associated with mitochondria. Bioenergetically, AC iHLCs are deficient in mitochondrial spare or reserve capacity, while the mitochondria of AC iHLCs display less fuel flexibility. In iHLCs and in human liver biopsies, PNPLA3 and MARC1, which are encoded by the two genes with the largest effect in AC-focused GWASs, are colocalized with LDs and mitochondria. Together, the transcriptomic pattern of oxidative stress gene activation and measures of oxidative stress in the AC iHLCs suggest that oxidative damage may be driving mitochondrial dysfunction and lipid accumulation in AC, thus contributing to its pathogenesis, which may be exacerbated by gene knockdown, as done here for *PNPLA3*, or partly reversed by in vitro treatments, for example with Aramchol.

## Methods

### Human samples

Venous blood for iHLCs was collected following informed consent from patient volunteers at VCU Medical Center or anonymous healthy donors at the NIH Clinical Center. The joint VCU/NIAAA human research protocol was approved as IRB FWA # 00005897 at VCU. All patients with alcohol cirrhosis (AC) were recruited at the VCU Medical Center and diagnosed after an intensive workup, including liver biopsy. Some demographics and clinical parameters of the patients with AC and the healthy (H) volunteers are shown in Table 1. Random patients with both gender were recruited with limited time frame with informed consent as per NIH guidelines.

Liver tissues (Liver Tissue Cell Distribution System, University of Minnesota, a NIH funded public resources, approved by University of Minnesotta) was obtained from six patients who underwent transplants for liver failure due to AC and three healthy controls. Liver tissues were obtained as paraffin block embedded. Liver Tissue Cell Distribution System, University of Minnesota was under NIH funding resource and provide liver tissues to all investigators.

### iHLCs

iHLCs were generated over 90 days via the five steps shown in the schematic diagram Fig. 1A. Peripheral blood mononuclear cells (PBMCs) were isolated from venous blood and stored frozen for future use. CD4 + T cells were isolated and activated from frozen PBMCs, with cells from H and AC groups always being processed together in batches. iPSCs were generated by infecting activated CD4 + T cells with CytoTune iPSC Sendai Reprogramming kit (# A16517, Thermo Fisher

Scientific) with four separate viruses to deliver the Yamanaka factors (hOct3/4; hSox 2; hKlf4 and hcMyc). iPSCs clones (3 positive clones from each individual) were expanded and pluripotency was confirmed. H or AC iPSClines simultaneously,iPSCS either being immediately differentiated to iHLCS s or stored as frozen and later differentiated to iHLCs. The above four steps were adapted from [reference][51]. iPSCs were differentiated to iHLCs generation using a published protocol[52].

### RNA-sequencing

RNA was isolated from four H and six AC iHLCs samples at day 90 or slightly later (Fig. 1A). RNA-sequencing was performed using an Illumina Novoseq 6000 sequencing system generating 50 million 150 bp paired-end poly-A captured total RNA reads per sample. Quality control, normalization and TPM (transcript per-kilobase million) computations were performed using the Ensembl reference genome for humans (GRCh38). Differential gene expression was detected using R Studio using Bioconductor and DESeq2 where cutoff values were kept at 1.5 fold increase or decrease. Correlation coefficients between samples were computed using the Pearson method in R studio. Gene set analyses based on gene-level statistics were performed against Wikipathways2019 with Piano (Bioconductor: https://bioconductor.org/packages/release/bioc/html/piano.html)[53]. Differentially expressed genes were color-coded (red = up, blue = down) and loaded onto the COLOR KEGG mapper to highlight gene expression differences in oxidative phosphorylation and the electron transport chain. Enrichment analyses showed 5278/8459 gene sets are upregulated in phenotype AC when analyzed with GSEA (Broad Institute, Inc., Massachusetts Institute of Technology) and oxidative stress-associated GSEA is shown in Fig S6.

### Real-time PCR for DGE

cDNA was prepared from total RNA from iHLCs using the High-Capacity cDNA Reverse Transcription kit (#4368813, Thermo Fisher Scientific). Quantitative RT-PCR was performed using the ABI 7500 Real-time PCR instrument. Each PCR reaction, for *CYP1A2*, *CYP2C9*, *PROX1*, *TBX3*, *AFP*, *ATT*, *ABCC2*, *MAOB*, *UGT1A1*, and *CYP3A* transcripts, was performed using Power SYBR™ Green PCR Master Mix (#4368708, Thermo Fisher) and primers (Qiagen) according to the manufacturer's recommendations.

### De novo lipogenesis

De novo lipogenesis or fatty acid synthesis was measured in iHLCs by incorporation of $^3H_2O$ into cellular fatty acids, as previously described[54].(PerkinElmer #NET001B001MC) Briefly, iHLCs were distributed into six-well plates and cultured for 24 h followed by 167 μCi $^3H_2O$ addition for 1 h. The reaction was terminated by the addition of chloroform/methanol (1:1). Fatty acids were extracted with ether and newly synthesized species were quantified by liquid scintillation spectrometry. The rate of lipogenesis was expressed as fold difference between H and AC iHLCs, using absolute nmol $^3H_2O$ incorporated into fatty acids per min per $10^8$ cells.

### Fatty acid uptake, triglycerides and phosphatidylcholine

Cell-based fluorometric assays were performed using the Free Fatty Acid Uptake Assay kit (#ab176768, Abcam). Briefly, iHLCs were grown in 96 well plate and incubate 1 h in serum-free media. Fatty acid dye-loading solution was added for 4 h and the fluorescence signal measured with a SpectraMax fluorescence microplate reader at Ex/Em = 485/515 nm.

Triglyceride content of iHLCs was measured using the Triglyceride Quantification Colorimetric/Fluorometric kit (# K622-100, Biovision). iHLCs($10^6$ cells) were suspended and lysed in 5% NP-40/ddH2O solution. The output was measured at OD 570 nm for colorimetric assay after mixing and incubating with assay reagents provided with the kit.

Phosphatidylcholine in iHLC cell lysates was measured using the Phosphatidylcholine Assay kit (#ab83377, Abcam) and a SpectraMax Microplate reader (Molecular Devices). Briefly, iHLCs were grown in 96 well plates, washed in PBS and suspended in assay buffer provided with the kit. Both standards and samples were incubated with kit components as directed and the measurement was performed at OD570 (OD is optical density).

## AI-driven workflow for lipid droplet analyses

3D Ariyascan images were analyzed using AIVIA[55]. Machine Learning based Pixel Classifier was applied to the lipid channel. Then the pixel classifier output was used as input for Cellpose 2.0 Deep learning-based object segmentation. "3D Labeled Object Segmentation" recipe was applied to segment the Cellpose result with Object diameter 0.25–50 $\mu$m, minimum Z depth of 4 planes, and a mesh smoothing factor of 1. Finally, the "Gate by Rules" sorted objects into three categories: Small Volume <1.0 $\mu$m$^3$, Medium Volume 1.0–4.0 $\mu$m$^3$ and Large volume >4.0 $\mu$m$^3$. LD size was normalized with machine learning nuclei counts (>1000) for each image.

## Mito tracker Deep Red staining

MitoTracker® Deep Red FM (#M22426, Invitrogen) is a far red-fluorescent dye (abs/em ~644/665 nm) that stains mitochondria and was used for mitochondrial localization in iHLCs. iHLCs were stained for 15 min at 100 nM with MitoTracker® Deep Red FM. Images were taken with a confocal microscope LSM 700 or 900 (Carl Zeiss) using oil- objectives (40×–60×) as described below.

## Measurement of real-time oxygen consumption and extra-cellular acidification rates

We used the eXF86 Seahorse Flux Bioanalyzer to measure real-time oxygen consumption rate (OCR) for mitochondrial respiration and extracellular acidification rate (ECAR) as a measure of glycolysis, assaying AC and H iHLC's simultaneously. OCRs and ECARs were measured using the Seahorse XF Cell Mito Stress Test Kit (#103708-100, Agilent) according to the manufacturer's instructions. Briefly, the cartridge plate was hydrated with XF calibrant buffer and incubated overnight (37 °C, CO$_2$-free). The assay medium with the full substrate (XF base DMEM medium #103575-100, Agilent) contained 1 mM pyruvate, 2 mM glutamine, and 10 mM glucose. The limited substrate medium was FAO Assay Medium composed of KHB buffer [111 mM NaCl, 4.7 mM KCl, 1.25 mM CaCl2, 2 mM MgSO4, 1.2 mM NaH2PO4] supplemented with 2.5 mM glucose, 0.5 mM carnitine, and 5 mM HEPES adjusted to pH 7.4 at 37 °C. These media were prepared immediately before use. Equal numbers of iPSCs at passages 6–7 were plated in 92 BD Matrigel-coated wells with four corner wells left empty as blanks. All cells were treated with growth factors for differentiation to iHLCs for 3–4 weeks. Cells were deprived of supplemental growth factors overnight before experiments. Cells were maintained for 1 h in either full substrate or limited substrate assay media before assays. Then the OCRs and ECARs were measured using a Flux analyzer. All wells in the plate were individually examined microscopically for cell clustering, uneven distribution or under 90% confluency of the surface was observed were excluded from analyses. The lack of difference in non-mitochondrial oxygen consumption indicates that the difference between AC and H iHLCs in terms of the OCR is specific to mitochondria.

For measurement of fuel dependency, an Agilent Seahorse XF Mito Fuel Flex Test kit (103260-100), an XF Long Chain Fatty Acid Oxidation Stress Test Kit (103672-100) and an XF Glutamine Oxidation Stress Test kit (103674-100) were used for the oxidation of the specific substrates FA and GLN. All methods were according to the manufacturer's recommendations and cells were processed as described above.

## Western Blotting of mitochondrial complex protein

Cell lysates were prepared in RIPA Lysis and Extraction Buffer (#89900, Thermo Fisher Scientific) with Halt™ Protease Inhibitor Cocktail (#78425, Thermo Scientific). Equal amounts of protein (100 ug) were loaded in a well of SDS-PAGE gel, transferred to a nitro-cellulose membrane subsequent to electrophoresis and probed with mitochondrial complex proteins followed by SuperSignal™ West Pico PLUS Chemiluminescent Substrate (#34577 Thermo Scientific). Antibodies used were Complex I(Abcam # ab109721), Complex II (abcam #ab109865), Complex III (ab109862) and complex V (ab109715). All antibodies were raised against human complexes and were used 1:1000 dilution.

## Confocal microscopy

All imaging was performed with either an LSM700 or LSM 900 (Carl Zeiss Microscopy) using 20× or 40–60× oil objectives. Images were processed with Zen software (Carl Zeiss Microscopy).

## Flow cytometry

For intracellular staining of albumin and HNF4a, iHLCs were fixed and permeabilized with Fixation/Permeabilization Kit (#554714, BD Bioscience) followed by blocking and 1-h staining with antibody to albumin or HNF4a (# ab106582 and ab92378, Abcam). After washing, cells were incubated with secondary Alexa Flur 488 and Alexa fluor 594 conjugated antibodies. Cells were washed and analyses were performed in a CytoFlex flow cytometer (Beckman Coulter).

For mitochondrial peroxidation, cells were incubated with Mito-PER (#ab146820, Abcam) in 12 well plates for 15 min according to the manufacturer's instructions. Cells were washed multiple times and detached with Versine (#A4239101, Thermofisher Scientific). Analytic flow cytometric analyses were performed on live cells with a CytoFlex flow cytometer (Beckman Coulter).

## Co-immunoprecipitation of 4-hydroxynenal proteins

For co-immunoprecipitation assays to measure 4-hydroxynenal modified proteins, protein A magnetic beads were equilibrated with RIPA buffer and pre-coated with 1% BSA. Immunoprecipitation assays were then performed using 500 $\mu$g of total protein. Clearing of cell lysates was done by the addition of 30 $\mu$l of 50% slurry of pre-coated protein A beads per 500 $\mu$g of protein. The supernatant was collected by centrifugation and the 4-HNE modified proteins were immunoprecipitated with anti-4-HNE monoclonal antibody (#HNE-J2, Genox Corporation) overnight at 4 °C. Fifty to 100 $\mu$l of 50% slurry were added to the mixture and incubated for 2 h at 4 °C with shaking. The beads were washed three times with RIPA buffer containing protease and phosphatase inhibitors. The immunoprecipitants were eluted by boiling for 5 min in an SDS sample buffer and subjected to SDS-PAGE analysis and Western blotting.

## SiRNA knock down of PNPLA3

Synthetic SMART selected siRNA duplexes for PNPLA3 target were purchased from Dharmacon RNA Technologies (Lafayette, CO). Functional siCONTROL non-targeting siRNA pool from Dharmacon was used as a negative control. Transfections of siRNA duplexes at 100–200 nM final concentration were carried out using DharmaFECT transfection reagents in iHLCs following manufacturer instructions. After transfection, cells were processed after 72 h for western blot or treatment. Viable cells were identified and counted by exclusion of Trypan Blue staining.

## AFP

AFP was quantified in IHLCs using Abcam's alpha Fetoprotein (AFP) Human in vitro ELISA (Enzyme-Linked Immunosorbent Assay) kit (#ab108838, Abcam, CA, USA) according to manufacturer's instruction. Briefly, iHLCs were rinsed with cold PBS and then scrape the cell

into a tube with 5 ml of cold PBS and 0.5 M EDTA. The suspension was centrifuged at 1500 rpm for 10 min at 4 °C and the supernatant was aspirated. The pellet was suspended in ice-cold Lysis Buffer (PBS,1% Triton X-100, Complete Mini EDTA-free protease inhibitor cocktail). Standards and samples were incubated with reagents supplied with the kit according to the direction and the sample were measures at OD 450.

### Urea secretion

Urea in the cell supernatant was measured using UREA assay kit (#MAK006Sigma -Aldrich) according to its instructions. Briefly, cells were suspended in ice-cold lysis buffer (PBS,1% Triton X-100, Complete Mini EDTA-free protease inhibitor cocktail). Urea concentration is measured by a coupled enzyme reaction, which results in a colorimetric (OD570 nm) product.

### Glycogen storage

Glycogen storage in iHLCs was quantified by glycogen assay kit (#ab65620, Abcam, CA, USA) and followed its protocol. Briefly, iHLCs (-$10^6$) were harvested after washing with cold PBS and suspended in 200 microliter of ddH$_2$O. The homogenate boiled for 10 min and centrifuged samples 10 min at 4 °C at 18,000 × $g$. Reaction mix were added as per instruction and the OD570 was measured for standard and samples.

### Determination of CYP450 activity

CYP450 activities in iHLCs were determined using Colorimetric Cytochrome P450 Reductase Activity Assay Kit (#ab204704Abcam, CA, USA) according to its instruction CYP450 activities in iHLCs were determined using Colorimetric Cytochrome P450 Reductase Activity Assay Kit (#ab204704 Abcam, CA, USA) according to its instruction. Briefly, a crude microsomal preparation for fresh iHLCs was extracted with the reagents and direction provided with the kit. After incubating with reagent mix provided with the kit, OD460 nm in kinetic mode was set up in a microplate reader. Activities were determined based on the formula provided from the manufacturer.

### Isolation of mitochondria from iHLCs

Mitochondria from iHLCs were isolated using Mitochondria Isolation Kit for Cultured Cells (#PI89874 Thermo scientific, CA, USA). Briefly, mitochondria were isolated on a reagent-based method with speed control vortex mix. Fresh iHLCs were processed for isolation of mitochondria.

### Mitochondrial super complex visualization using native PAGE

Isolated mitochondrial from fresh iHLCs were dissolved mitochondrial isolation buffer (3 mM HEPES-KOH pH 7.4, 210 mM mannitol, 70 mM sucrose, 0.2 mM EGTA, Complete Mini EDTA-free protease inhibitor cocktail) with 5 mg/ml digitonin as described[56]. Equal amount of 50 μg protein from six healthy and six AC patient-derived iHLCs were run on a stain-free 4–15% TGX gel (Biorad) with Tris-glycine buffer and stained with Simplyblue Safe protein staining reagents (Invitrogen). Unstained protein marker Native Mark (Invitrogen #LC0725) was used as molecular weight marker.

### Measurement of mitochondrial DNA/ nuclear DNA levels

Mitochondrial DNA was isolated using Mitochondrial DNA Isolation Kit (#ab65321Abcam, CA, USA) We also measured the amount of mtDNA by quantitative PCR using the relative levels of mtDNA-encoded gene ND2 and the nuclear gene, GAPDH. The primers used are

 MT-ND4: GGACTCCACTTATGACTCCC
 MT-ND4: GGTTGAGAATGAGTGTGAGGC
 GAPDH (Qiagen #QT00079247)

### Mitochondrial reduced and total glutathione

Reduced and total glutathione from mitochondrial extracts were determined using flurometric GSH/GSSG assay kit (#ab138881, Abcam CA, USA). iHLCs (-$10^7$ cells) were harvested and washed with cold PBS. Cells were lysed with lysis buffer provided with the kit and centrifuged to remove any insoluble materials. The supernatant from samples and standards was incubated with assay mixtures for GSH and total glutathione assays. The fluorescence at Ex/Em = 490/520 nm with a fluorescence microplate reader.

### PARP1 enzyme activity

Poly-ADP-ribosylation of proteins is a posttranslational event in response to cellular stress. PARP1 activity was measured in iHLCs using HT universal colorimetric PARP Assay kit (#4677-096 K Biotechnique, MN, USA) according to its instructions. IHLCs were lysed in lysis buffer (PBS,1% Triton X-100, Complete Mini EDTA-free protease inhibitor cocktail). After rehydrating 8 well strips provided by manufacturer, cell lysates and assay reagents were added according to the protocol provided. The absorbance at 450 nm were read for the plate.

### DNA fragmentation

Cellular stress marker DNA fragmentations in iHLCs were determined using Cell death ELISA (#11544675001 Roche, Sigma -Aldrich, MO, USA). Briefly, iHLCs (-$5 \times 10^4$ cells) were suspended in the incubation buffer provided with the kit and centrifuged at 20,000 $g$ for 10 min. After activating the coated plate provided from the manufacturer, 100 μl of samples lysates were added to each well. After adding reagent, the absorbance was measured at 405 nm.

### Statistical analyses

All values are represented as mean ± SEM. Statistical analysis of the data was performed either by unpaired Student's $t$ test or one-way analysis of variance followed by Tukey's post hoc test for multiple comparisons. Analyses were conducted using GraphPad-Prism 8.01 software. Significant $P$ values were $P < 0.05$. Experiments were done in blinded fashion, and all received samples were processed without any subsequent exclusion.

### Illustrated figures

The part of Figs. 4f, 5b, 7d, 8f. was obtained from Servier Medical Art by Servier, licensed under a Creative Commons Attribution 3.0 Unported License.

### Reporting summary

Further information on research design is available in the Nature Portfolio Reporting Summary linked to this article.

## Data availability

Source data are provided with this paper. RNAseq data is available through GEO accession number GSE254610. Any additional information is available upon request to the corresponding author (David Goldman, davidgoldman@mail.nih.gov) Source data are provided with this paper.

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

## Acknowledgements

This work was supported by intramural funds from the National Institute on Alcohol Abuse and Alcoholism to D.G. (ZIA AA000301, ZIA AA000421) and with partial support from the Sanyal-Stravitz Center, Virginia Commonwealth University].

## Author contributions

B.M. and D.G. conceived the project and designed the experiments. B.M. and D.G. analyzed the data and wrote the paper. B.M., D.G., A.S. and G.K. edited the manuscript. B.M., C.M., P.H.,. M.M., C.H., E.W., Q.Y., D.M. conducted experiments and analyzed the data. A.S., A.O., F.M. provided clinical samples and analyzed associated data.

## Funding

## Competing interests

The authors do not have any financial and non-financial interest in design or preparation of the manuscript. D.M. receives a salary from Leica Microsystems where he is Manager for Aivia Software, however received no compensation to assist for this manuscript. A.J.S. have stock options in Genfit, Akarna, Tiziana, Durect Inversago, and Galmed. He has served as a consultant to Astra Zeneca, Salix, Tobira, Takeda, Jannsen, Gilead, Terns, Birdrock, Merck, Valeant, Boehringer Ingelheim, Bristol Myers Squibb, Lilly, Hemoshear, Novartis, Novo Nordisk, Pfizer, 89 bio, Regeneron, Alnylam, Akero, Tern, Histoindex, Corcept, Poxel, Path AI, and Genfit. His institution has received grant support from Gilead, Salix, Tobira, Bristol Myers, Shire, Intercept, Merck, Astra Zeneca, Malinckrodt and Novartis. He receives royalties from Elsevier and UptoDate. However, A.J.S. received no compensation from any source for this manuscript.
