## [Peer Review File · Nature Communications]

A patient-based iPSC-derived hepatocyte model of alcohol-associated cirrhosis reveals bioenergetic insights into disease pathogenesis.REVIEWER COMMENTS

Reviewer #1 (Remarks to the Author):

In this manuscript the authors employed iPSC-derived hepatocytes (iHLCs) from patients with AC to hypothesize that hepatocyte-intrinsic genetic and bioenergetic factors leading to AC would be preserved in such cells.

In an attempt to address their hypothesis, they employed transcriptome, bioenergetics and histological measurements and show that AC iHLCs had a greater number of lipid droplets (LDs) and LD-associated mitochondria compared to control cells. These pre-pathologic indicators were effectively reversed by aramchol, an inhibitor of stearyl-CoA desaturase.

The expression of MARC1 and PNPLA3-genes implicated by GWAS in non-alcoholic fatty liver disease, correlated with LD-associated mitochondria-mediated oxidative damage in AC iHLCs, and knockdown of PNPLA3 expression exacerbated mitochondrial deficits. Furthermore, AI-aided machine learning analysis of LD volume revealed increases in large and medium LDs after PNPLA3 knockdown, whereas small LDs decreased in number.

Based on these findings they propose that “differences in mitochondrial bioenergetics and LD formation are intrinsic to AC hepatocytes and can play a role in the pathogenesis of AC”.

Comments

Though an interesting study there are numerous shortfalls;

1. They hypothesize that hepatocyte-intrinsic genetic and bioenergetic factors leading to AC would be preserved in these iHLCs- where is the evidence in support of this?, they have not screened for NALFD and AC associated variants/ mutations in for example PNPLA3, MARC1, FAF2, HSD17B13, HNF1A, and SERPINA1. There seems to be a misconception here, the authors have not considered the fact that any epigenetic marks in the patient somatic cells are erased upon iPSC derivation.
2. The description of the individuals- both H and AC from which iPSCs were derived (Table 1) is wholly inadequate- they should list all individuals, age and gender and of course the associated clinical parameters
3. Characterisation of the iPSC differentiated iHLCs- it is normal practice to analyse glycogen storage, urea production and CYP-450 activity with reference to the golden standard- primary hepatocytes
4. In the materials and methods section the authors should be explicit on the number of iPSCs differentiated into iHLCs- both from H and AC individuals
5. They claim the iHLCs are mature but they have not demonstrated this- they have to analyse AFP expression and additional maturation markers- Albumin alone is not enough or convincing.
6. It is not clear if the H-iHLCs and AC-iHLCs were stressed prior to carrying out the various analyses- bioenergetics and lipid droplets (LDs) formation
7. Bioinformatics is weak. They have to compare the transcriptomes of the various iHLCs to that of primary liver-biopsy derived mature hepatocytes
Just focusing on the 343 statistically significant DEGS is not enough (what was the cut-off ratio?). of interest will be the geneset expressed exclusively in AC-iHLCs and H-iHLCs and their associated GOs and pathways.

8. The authors state- “To understand the physiological function of PNPLA3 in iHLCs, we knocked down PNPLA3 expression via siRNA. PNPLA3 knock down lead to significantly larger LD size in AC

iHLC compared to scrambled siRNA-treated AC iHLCs”

However, there is no description in the M&M section on how the SiRNA experiment was carried out, the efficiency and also a PCR and western blot analyses demonstrating the knockdown in AC and H iHLCs. PLIN2 expression is also associated with LD formation- did they analyse this?

9. The data provided does not support the conclusions drawn

Reviewer #2 (Remarks to the Author):

The authors performed this translational clinical research study using the induced pluripotent stem cell-derived hepatocytes (iHLCs) from patients with AC to examine whether hepatocyte-intrinsic genetic and bioenergetic factors leading to AC are preserved in such cells. The authors found that AC iHLCs had a greater number of lipid droplets (LDs) and LD-associated mitochondria compared to control cells, which were effectively reversed by aramchol, an inhibitor of stearyl-CoA desaturase. Bioenergetically, AC iHLCs had lower spare capacity and slower ATP production. Their uptake of fatty acids, triglycerides and phosphatidylcholine was increased, and de novo lipogenesis trended higher, while metabolic flexibilities of AC iHLC mitochondria for fatty acids and glutamate were weakened. MARC1 and PNPLA3, genes correlated with LD-associated mitochondria-mediated oxidative damage in AC iHLCs, and knockdown of PNPLA3 expression exacerbated mitochondrial deficits. These findings suggest that differences in mitochondrial bioenergetics and LD formation are intrinsic to AC hepatocytes and can play a role in the pathogenesis of AC.

The authors need to be congratulated for their study with this novel study using pluripotent cells iHLCs to study pathogenesis of AC, which is clearly a strength of the study. However, there are several minor concerns to this reviewer which authors may like to address:

1. As cirrhosis is an inflammatory state, were there any changes in the cytokine milieu (both pro- and anti-inflammatory cytokines). If cannot be examined now, this may be included in the discussion as a limitation of their study and propose to examine in future studies.
2. Although authors refer to the technique on generating iHLCs (Ref. 51), the authors may allude to this in the methods section, especially how the biopsies from healthy controls and tissue from explants were used for this purpose.
3. As the non-mitochondrial OCR is secondary to pro-oxidant and pro-inflammatory enzymes especially cytochromes, how do the authors explain these to be similar if the main pathology is oxidative stress. Does glycolysis compensates for this and if this component of bioenergetics is measured by the authors?
4. The finding of abnormalities in the OCR with normal mitochondrial complexes suggests that there is inherent defect in the hepatocytes to generate ATP from mitochondria without decrease in the mitochondrial mass, how about mitochondrial DNA changes and abnormalities and mitochondrial proteins differences in the two populations.
5. Were any measurements made for oxidative burst of the hepatocytes. This is important as this oxidative stress is the main driving force for mitochondrial abnormalities as per the findings of the study.
6. There has been suggestion to incorporate various components of bioenergetics profile of an individual to provide a composite number or bioenergetics health index or BHI. The authors may like

to examine this and see if the AC hepatocytes vs. healthy controls have a lower BHI. For this other measurements especially proton leak associated OCR will need to be incorporated (PMID 24895057). 7. Clearly, study findings are novel and of great premise to move this field forwards both in understanding as well as treatment options as authors show using aramchol in their experiments and some other studies (PMID 36788015). However, these studies to this reviewer do not explain susceptibility to alcohol associated cirrhosis in only 10-20% of at risk individuals. Specifically, were healthy controls active drinkers without liver disease?

In addition, there are some typos in the manuscript. For example, in the introduction first page line 77, font size is different.

Reviewer #3 (Remarks to the Author):

In the present study, Mukhopadhyay et al, set up a patient-based cell system approach to model alcohol associated cirrhosis focusing on bioenergetic and lipid droplet homeostasis. Using an induced pluripotent stem cell-derived hepatocyte system (iHLC) from patients with alcoholic cirrhosis, the authors examine the profile of mitochondrial respiration and oxidative phosphorylation and morphology of lipid droplets compared to iHLCs from normal subjects. iHLCs from cirrhotic patients exhibit increased population of lipid droplet generation associated with mitochondria, while extracellular flux analyses demonstrate decreased oxidative phosphorylation activity compared to control iHLCs. Inhibition of stearoyl CoA desaturase by aramchol decreases the genesis of lipid droplets and enhances bioenergetic activity of mitochondria. Moreover, AC iHLC display increased expression of MARC1 and PNPLA3, two genes associated with metabolic liver disease, in association with mitochondria, and the knockdown of PNPLA3 increased the size of lipid droplets. The findings suggest that differences in mitochondrial bioenergetics and lipid droplet formation are innate features of hepatocytes in alcoholic cirrhosis and that can play a role in ALD pathogenesis.

COMMENTS

1. Although the approach used is novel and of potential interest, the findings reported are totally expected in the sense that the changes in HLCs from cirrhotic patients reflect the known deficits described in the disease. Many papers have described that mitochondrial function from patients with ALD are impaired or defective with lower oxidative phosphorylation potential. The findings described by Mukhopadhyay et al are confirmatory of these results, using a sophisticated cell system.
2. The limitation with this kind of investigation is its correlative nature. The approach and design of the study do not allow to extract cause-and-effect consequences and whether the deficits of mitochondrial function is the cause or the consequence of the disease.
3. Although lipid droplets and mitochondria are known to associate, the authors do not provide mechanistic explanation as to what causes the impairment in oxidative phosphorylation in alcoholic cirrhosis. Authors examine the expression levels of complexes I, II, III and V but not determine whether the assembly of respiratory supercomplexes is defective. This approach would be of greater significance as the respirasomes control oxidative phosphorylation.

4. Findings in Fig 8D indicates that iHLCs from AC patients exhibit lipid peroxidation. Authors did not examine the status of the mitochondrial antioxidant strategies.

5. The approach establish here can be useful and should be complementary of additional strategies. BY itself this model precludes the estimation of the relationship between the hepatocellular dysfunction, as described here, and other critical players of the diseases such nonparenchymal cells (HSC, Kupffer cells) involved in fibrosis and inflammation.

6. Surprisingly, the manuscript is poorly written with sentences needing a major revision (e.g. Introduction sentence in line 59). Also paragraphs in line 68-71 and 73-77 are duplicated.

7. Nomenclature for HiHLCs or CiHLCs should be homogenized.

Responses to REVIEWER COMMENTS

We sincerely thank the reviewers for their several apt comments and suggestions to improve our paper. In response, we have thoroughly revised the paper to address each of the issues that were raised, and as addressed here point by point. We feel that the resulting paper is substantially improved, corrects each of the defects noted. Thereby, it represents a substantial contribution to Nature Communications, describing as it does impactful findings on the preservation of histologic, transcriptomic, and metabolic differences in induced Hepatocyte-like Cells derived from patients with Alcohol Cirrhosis, and - as now added- showing partial reversal via the drug aramchol and exacerbation via inhibition of PNPLA3.

Reviewer #1 (Remarks to the Author):

In this manuscript the authors employed iPSC-derived hepatocytes (iHLCs) from patients with AC to hypothesize that hepatocyte-intrinsic genetic and bioenergetic factors leading to AC would be preserved in such cells.

In an attempt to address their hypothesis, they employed transcriptome, bioenergetics and histological measurements and show that AC iHLCs had a greater number of lipid droplets (LDs) and LD-associated mitochondria compared to control cells. These pre-pathologic indicators were effectively reversed by aramchol, an inhibitor of stearyl-CoA desaturase.

The expression of MARC1 and PNPLA3-genes implicated by GWAS in non-alcoholic fatty liver disease, correlated with LD-associated mitochondria-mediated oxidative damage in AC iHLCs, and knockdown of PNPLA3 expression exacerbated mitochondrial deficits. Furthermore, AI-aided machine learning analysis of LD volume revealed increases in large and medium LDs after PNPLA3 knockdown, whereas small LDs decreased in number.

Based on these findings they propose that “differences in mitochondrial bioenergetics and LD formation are intrinsic to AC hepatocytes and can play a role in the pathogenesis of AC”.

-We sincerely appreciate your critical reading of our manuscript.

Comments

Though an interesting study there are numerous shortfalls;

1. They hypothesize that hepatocyte-intrinsic genetic and bioenergetic factors leading to AC would be preserved in these iHLCs- where is the evidence in support of this?, they have not screened for NALFD and AC associated variants/ mutations in for example PNPLA3, MARC1, FAF2, HSD17B13, HNF1A, and SERPINA1. There seems to be a misconception here, the authors have not considered the fact that any epigenetic marks in the patient somatic cells are erased upon iPSC derivation.

-Thank you for raising this critical and very thoughtful point. Large scale GWAS, including the GWAS in which we have been directly involved, have demonstrated the important roles of the variants this reviewer mentions as single SNPs or in combination with others. However, we feel it is beyond the scope of the present study to understand their individual or combinatorial role. Even so, we are intensely interested in their role and will be the focus of future studies. Such investigations will require genome engineering to understand the critical or specific roles of individual variants, making this limitation beyond the scope the current study. Also, it should be noted that such studies may be

feasible, but nevertheless challenging, for polygenic combinations that currently are insufficiently understood. However, in response to this comment, we targeted PNPLA3, the top hit in AC GWAS, showing that RNA interference of PNPLA3 leads to increased lipid droplet size. Furthermore, in the present revision we have taken pains to clarify two issues: 1) several of the top hits in AC GWAS are lipid droplet-related genes that are expressed in hepatocytes, thereby implicating hepatocytes as a critical cellular context in which to identify intrinsic features associated with the risk of AC; 2) the presently available polygenic scores, or gene scores, are not highly predictive of AC. Therefore, the best indicator that an iHLC represents a hepatocyte that is genetically predisposed for AC is not its genotype per se, but that the iHLC is derived from a patient with AC, as we have done here.

Here, to respond more fully to the reviewer's critique, we want to add that following the usual pattern of gene discovery in complex diseases, over time, and with larger samples, more of the genetic variance in AC vulnerability will be explained. However, the effect sizes of the new loci detectable only in very large datasets is likely to be very small individually, and they may be expected to increase risk when found in polygenic combinations. Therefore, although we will measure AC polygenic scores in iHLCs as the polygenic scores become more informative, we believe that it will remain critical to study iHLCs derived from patients with proven AC, as only a fraction of heavy drinkers ever develop AC.

2. The description of the individuals- both H and AC from which iPSCs were derived (Table 1) is wholly inadequate- they should list all individuals, age and gender and of course the associated clinical parameters

-Thank you for your excellent suggestion. We have modified the table accordingly.

3. Characterisation of the iPSC differentiated iHLCs- it is normal practice to analyse glycogen storage, urea production and CYP-450 activity with reference to the golden standard- primary hepatocytes

-We agree and have tested these critical parameters previously. However, we have now repeated those experiments in our iHLCs AND in human primary hepatocytes. These data verify that these aspects of hepatocyte function are intact in iHLCs are presented in Fig Suppl 2 and the Results, and the Methods are cited for each of these assays.

4. In the materials and methods section the authors should be explicit on the number of iPSCs differentiated into iHLCs- both from H and AC individual.

-We have modified the Methods section to clarify this.

5. They claim the iHLCs are mature but they have not demonstrated this- they have to analyse AFP expression and additional maturation markers- Albumin alone is not enough or convincing.

-As suggested, we have included multiple maturation markers and compared them between day 1 and day 21. The figure is provided as Supplemental Figure 1. It is also important to note that whereas the iHLCs are indeed hepatocyte-like in histologic, transcriptomic, metabolic and other functional aspects, they still bear some signs of immaturity; namely, expression of AFP, CYP3A, ATT, ABCC2, MAOB and UGT1A1. Therefore, we have now taken pains to emphasize in the paper that iHLCs represent a model

of young hepatocytes that still manifest features not seen in fully mature, or – as might be said- older hepatocytes.

6. It is not clear if the H-iHLCs and AC-iHLCs were stressed prior to carrying out the various analyses- bioenergetics and lipid droplets (LDs) formation

-These cells were not stressed during or before experiments. We also measured cell death markers at Day 1 and Day 21 finding no significant changes during the course of differentiation (Fig Suppl) or signs that stress was induced. However, we acknowledge that the unnatural conditions of cell culture in the absence of supporting cells and structure as in the liver could itself represent a condition essential to the manifestation of histologic, metabolic and transcriptomic differences in AC iHLCs. Indeed, in vivo hepatocytes do not ordinarily accumulate large fat droplets except after alcohol or a metabolic challenge, such as a high fat diet. In vitro, we cultured AC and H iHLCs under identical conditions, but the AC iHLCs responded differently to those conditions. Rather than probing the effects of variations in cell culture conditions, we chose to perform more specific and mechanistically informative intervention;, namely, by inhibiting PNPLA3 and treating the cells with Aramchol . In the future, the variety and number of interventions that would be informative in these cells is practically unlimited, enabling specific aspects of cell nutrition, oxidative stress and other systems to be targeted.

7. Bioinformatics is weak. They have to compare the transcriptomes of the various iHLCs to that of primary liver-biopsy derived mature hepatocytes

Just focusing on the 343 statistically significant DEGs is not enough (what was the cut-off ratio?). of interest will be the geneset expressed exclusively in AC-iHLCs and H-iHLCs and their associated GOs and pathways.

-The cut-off ratios are now given for the DEGs, as requested. Furthermore, we discuss genes and pathways intrinsic to hepatocytes and were implicated by the GO and GSEA analyses of the DEGs, and together with metabolic and oxidative damage findings that mesh with these transcriptome findings. Regarding DEGs, here and in the future we and others will follow up on other gene networks implicated and using probes targeting those pathways. Here, we focused mainly on LDs and bioenergetics in the development of AC.

Thank you also for the interesting suggestion to compare our findings in iHLCs to primary hepatocytes. We believe this is beyond the scope of our study. We note, and were relieved, that the reviewer did not suggest a comparison of AC and H iHLC to AC and H primary hepatocytes. Like the reviewer, we think, we believe the core issue is the relevance of iHLCs or primary hepatocytes to hepatocytes resident in the liver. Our publicly available transcriptome data on AC and H iHLCs, introduced by this paper, will represent exactly that sort of comparative data resource as people move forward with comparative studies of liver transcriptome in primary hepatocytes cultured in different ways, liver single-cell transcriptomics, and spatial transcriptomics. We note that some data have been published based on primary liver biopsy-derived mature hepatocytes, but we were not able to derive primary hepatocytes from liver biopsies of patients with cirrhosis. Validation of iHLCs as a hepatocyte model, as we and others have done, is critical. Here, we have been careful to evaluate the authenticity of the iHLC's as a hepatocyte model using transcriptome features, but also histologically, metabolically and by other aspects of function. However, we believe that a parallel comparison of hepatocytes derived from AC and H livers is a task that would be difficult to execute technically and that is unwarranted. We recognize that culture of primary hepatocytes can be accomplished in several ways and has been

done for more than two decades via methods that encourage hepatocyte survival and proliferation. A comparative study could involve the investigation of similarities and differences between iHLCs and primary hepatocytes at different timepoints and under different cell culture conditions, as actually implied by this reviewer's earlier comment about the potential effect of cell culture conditions on expression of phenotypes by the iHLCs. By way of contrast, the iHLCs represent a more consistent and fungible reagent, a variety of experiments being feasible over time in iHLCs derived from the same patient, and thus with better opportunities for replication, validation and extension of results than if experiments are carried out on primary hepatocytes made on a one-time basis from patients with unique genetic backgrounds and clinical histories.

8. The authors state- "To understand the physiological function of PNPLA3 in iHLCs, we knocked down PNPLA3 expression via siRNA. PNPLA3 knock down lead to significantly larger LD size in AC iHLC compared to scrambled siRNA-treated AC iHLCs"
However, there is no description in the M&M section on how the SiRNA experiment was carried out, the efficiency and also a PCR and western blot analyses demonstrating the knockdown in AC and H iHLCs. PLIN2 expression is also associated with LD formation- did they analyse this?

-Thank you very much for pointing out this omission. We have now added these validating analyses to the revised paper.

9. The data provided does not support the conclusions drawn

-We appreciate the reviewer's concern in this regard, but with the new additional data based on the comments of all three referees we feel the revised paper does indeed have the necessary data to support the conclusions drawn. We hope the reviewer agrees with our assessment.

Reviewer #2 (Remarks to the Author):

The authors performed this translational clinical research study using the induced pluripotent stem cell-derived hepatocytes (iHLCs) from patients with AC to examine whether hepatocyte-intrinsic genetic and bioenergetic factors leading to AC are preserved in such cells. The authors found that AC iHLCs had a greater number of lipid droplets (LDs) and LD-associated mitochondria compared to control cells, which were effectively reversed by aramchol, an inhibitor of stearoyl-CoA desaturase. Bioenergetically, AC iHLCs had lower spare capacity and slower ATP production. Their uptake of fatty acids, triglycerides and phosphatidylcholine was increased, and de novo lipogenesis trended higher, while metabolic flexibilities of AC iHLC mitochondria for fatty acids and glutamate were weakened. MARC1 and PNPLA3, genes correlated with LD-associated mitochondria-mediated oxidative damage in AC iHLCs, and knockdown of PNPLA3 expression exacerbated mitochondrial deficits. These findings suggest that differences in mitochondrial bioenergetics and LD formation are intrinsic to AC hepatocytes and can play a role in the pathogenesis of AC.

The authors need to be congratulated for their study with this novel study using pluripotent cells iHLCs to study pathogenesis of AC, which is clearly a strength of the study.

-We are very thankful for the reviewer's support and constructive comments.

However, there are several minor concerns to this reviewer which authors may like to address:

1. As cirrhosis is an inflammatory state, were there any changes in the cytokine milieu (both pro- and anti-inflammatory cytokines). If cannot be examined now, this may be included in the discussion as a limitation of their study and propose to examine in future studies.

-We completely agree with this comment, but we could not examine the inflammatory milieu as suggested, but this will be a point for future studies.

2. Although authors refer to the technique on generating iHLCs (Ref. 51), the authors may allude to this in the methods section, especially how the biopsies from healthy controls and tissue from explants were used for this purpose.

-We have modified the Methods section accordingly. Thank you.

3. As the non-mitochondrial OCR is secondary to pro-oxidant and pro-inflammatory enzymes especially cytochromes, how do the authors explain these to be similar if the main pathology is oxidative stress. Does glycolysis compensates for this and if this component of bioenergetics is measured by the authors?

-We measured both components in H and AC iHLC groups. Glycolysis contributed to <10% of ATP production, which still may contribute to the energetics of the cells. Regarding alternative source(s) of ROS, we observed, and report, a trend for reduced mitochondrial complex activities in AC iHLCs compared to H iHLCs. This may directly contribute superoxide generation. We have now discussed this possibility in the revised Discussion although further investigation will be required to identify the source of enhanced ROS that we observed in the AC iHLCs.

4. The finding of abnormalities in the OCR with normal mitochondrial complexes suggests that there is inherent defect in the hepatocytes to generate ATP from mitochondria without decrease in the mitochondrial mass, how about mitochondrial DNA changes and abnormalities and mitochondrial proteins differences in the two populations.

-We greatly appreciate this valuable comment. Mitochondrial DNA/abnormalities are significantly reversed by aramchol treatment, supporting the notion that they have a limited contribution to mitochondrial metabolic impairment in AC iHLCs. We measured mtDNA/nDNA as suggested by the editor based on your comment, and we did not see any significant changes (Fig Suppl). We also provided a detailed discussion on this point in the revised paper.

5. Were any measurements made for oxidative burst of the hepatocytes. This is important as this oxidative stress is the main driving force for mitochondrial abnormalities as per the findings of the study.

-We have not come across any standard method for measuring oxidative burst in hepatocytes. Such assays are mainly performed in immune cells which are non-adherent. Here we face a technical challenge to pursue this interesting idea because hepatocytes are adherent and viability is

significantly lost after detachment. All such assays require the use of a flow cytometer but marked clogging due to cell death was observed during the treatment. These challenges have so far resulted in inconsistent data but because of this comment our interest has been aroused and we will continue to seek methods to more dynamically measure functions in hepatocytes, and at the single cell level where functional variations may be most salient.

6. There has been suggestion to incorporate various components of bioenergetics profile of an individual to provide a composite number or bioenergetics health index or BHI. The authors may like to examine this and see if the AC hepatocytes vs. healthy controls have a lower BHI. For this other measurements especially proton leak associated OCR will need to be incorporated (PMID 24895057).

-We generated the BHI index, which is included as a supplementary figure. This is a very important suggestion, because our observation is that the BHI formula as now constructed might be usefully revised, or there might be more than one formula to capture different aspects of bioenergetic health.

7. Clearly, study findings are novel and of great premise to move this field forwards both in understanding as well as treatment options as authors show using aramchol in their experiments and some other studies (PMID 36788015). However, these studies to this reviewer do not explain susceptibility to alcohol associated cirrhosis in only 10-20% of at risk individuals. Specifically, were healthy controls active drinkers without liver disease?

-Thank you again for this praise and valuable point. It is not known why only 10-20% of at-risk individuals develop liver cirrhosis. We are improving our ability to address this important question in our ongoing iHLC project on heavy drinkers with or without alcohol liver cirrhosis. Our healthy controls were active social drinkers but not heavy drinkers. Potentially, lifelong heavy drinkers without AC could be a more powerful group in which to detect protective mechanisms. However, the relatively small percentage of heavy drinkers who develop AC implies that our unscreened healthy controls are an effective contrast group to AC, and as they proved to be. We are now creating a new IRB protocol and will follow up after this study. This is very important question and we have added a discussion point to the revised paper accordingly.

In addition, there are some typos in the manuscript. For example, in the introduction first page line 77, font size is different.

-We are sorry for those inadvertent mistakes. They were addressed in the revision, where we have made extensive corrections.

Reviewer #3 (Remarks to the Author):

In the present study, Mukhopadhyay et al, set up a patient-based cell system approach to model alcohol associated cirrhosis focusing on bioenergetic and lipid droplet homeostasis. Using an induced pluripotent stem cell-derived hepatocyte system (iHLC) from patients with alcoholic cirrhosis, the authors examine the profile of mitochondrial respiration and oxidative phosphorylation and morphology

of lipid droplets compared to iHLCs from normal subjects. iHLCs from cirrhotic patients exhibit increased population of lipid droplet generation associated with mitochondria, while extracellular flux analyses demonstrate decreased oxidative phosphorylation activity compared to control iHLCs. Inhibition of stearoyl CoA desaturase by aramchol decreases the genesis of lipid droplets and enhances bioenergetic activity of mitochondria. Moreover, AC iHLC display increased expression of MARC1 and PNPLA3, two genes associated with metabolic liver disease, in association with mitochondria, and the knockdown of PNPLA3 increased the size of lipid droplets. The findings suggest that differences in mitochondrial bioenergetics and lipid droplet formation are innate features of hepatocytes in alcoholic cirrhosis and that can play a role in ALD pathogenesis.

COMMENTS

1. Although the approach used is novel and of potential interest, the findings reported are totally expected in the sense that the changes in HLCs from cirrhotic patients reflect the known deficits described in the disease. Many papers have described that mitochondrial function from patients with ALD are impaired or defective with lower oxidative phosphorylation potential. The findings described by Mukhopadhyay et al are confirmatory of these results, using a sophisticated cell system.

-We agree with the reviewer. However, our study represents a sufficient advance to be published in this journal, in our estimation, because it provides data showing that there is a significant correlation of mitochondrial-associated lipid droplets with oxidative phosphorylation, which can be reversed by the approved drug aramchol. Furthermore, this study establishes AC iHLCs as an in vitro model derived from any iPSC line and that can be used reagent-like in multiple experiments, an even by multiple investigators, thus breaking a critical technical barrier to progress in understanding AC and other liver diseases in which hepatocytes play critical roles.

2. The limitation with this kind of investigation is its correlative nature. The approach and design of the study do not allow to extract cause-and-effect consequences and whether the deficits of mitochondrial function is the cause or the consequence of the disease.

-We agree with this point of view. However, the partial reversal of the phenotype by Aramchol treatment indicates the important role of mitochondrial-associated lipid droplets in the defects of mitochondrial function in our patient-derived cell-based system.

3. Although lipid droplets and mitochondria are known to associate, the authors do not provide mechanistic explanation as to what causes the impairment in oxidative phosphorylation in alcoholic cirrhosis. Authors examine the expression levels of complexes I, II, III and V but not determine whether the assembly of respiratory supercomplexes is defective. This approach would be of greater significance as the respirasomes control oxidative phosphorylation.

-Thank you for pointing out this limitation. We have now determined mitochondrial complex activities and Complex II and complex IV have clear trends for reduced activities in AC iHLCs compared to H iHLCs, which may account for one of the sources of oxidative stress. We also measured mitochondrial reduced glutathione levels, which were lower in AC iHLCs. These can be main contributors to the generation of oxidative stress during AC progression, and as can be further probed in iHLC.

4. Findings in Fig 8D indicates that iHLCs from AC patients exhibit lipid peroxidation. Authors did not examine the status of the mitochondrial antioxidant strategies.

-Thank you for your suggestion. We performed new experiments to address this point and observed a reduced trend in mitochondrial antioxidant defense using mitochondrial fractions to measure the reduced glutathione-to-total glutathione status.

5. The approach establish here can be useful and should be complementary of additional strategies. BY itself this model precludes the estimation of the relationship between the hepatocellular dysfunction, as described here, and other critical players of the diseases such nonparenchymal cells (HSC, Kupffer cells) involved in fibrosis and inflammation.

-We completely agree with the reviewer about the potential role of nonparenchymal cells, such as HSCs and Kupffer cells, in fibrosis and inflammation. To assess such a complex inter-cellular communication a well-developed, multicellular-based organoid system would be useful. However, we can not feasibly develop such a system in a reasonable timeframe during the revision process of this paper. Furthermore, such heterocellular system will introduce other complexities, and for example making it more difficult to attribute a functional difference to a particular cell, or to the composition and method of production of the organoid. Thus, we will explore such a system in a future study, but we have added this point to the Discussion, emphasizing the complementarity of different models.

6. Surprisingly, the manuscript is poorly written with sentences needing a major revision (e.g. Introduction sentence in line 59). Also paragraphs in line 68-71 and 73-77 are duplicated.

-We are sorry for these issues. We have corrected them in the extensively revised manuscript.

7. Nomenclature for HiHLCs or CiHLCs should be homogenized.

-Thank you. We now use the nomenclature H iHLC / AC iHLC throughout. Our major reason for using the AC iHLC nomenclature is that we are also working with NASH-associated cirrhosis, which may conflict with the CiHLC nomenclature.

REVIEWER COMMENTS

Reviewer #1 (Remarks to the Author):

I am satisfied with the revised manuscript

Reviewer #2 (Remarks to the Author):

The authors have appropriately addressed all the comments and this reviewer does not have any further comments to make. It will be an important addition to the literature and once again this reviewer would like to thank and congratulate the authors for their work.

Reviewer #3 (Remarks to the Author):

In this revised version of original manuscript, authors addressed many of the critiques raised and concerns noted. There is still one point that is unresolved, which is related to the reason why the AC iHLCs appear to be intrinsically impaired in their OCR, including the oxygen consumption directed to synthesis of ATP as shown in Fig 4. The attempt of the authors to explain this in the Suppl Figure 8 is clearly insufficient as the activities of complex I, II and IV do not help in elucidating the phenotype of the mitochondrial function of AC iHLCs. Although the activities of these individual complexes measured do not change, mitochondrial complexes are organized in supercomplexes structures responsible for cellular respiration, which may be altered in the AC iHLCs. This is what was indicated in the first round of revision and was not explored in the revised version.

Reviewer #1 (Remarks to the Author):

I am satisfied with the revised manuscript

-Thank you very much.

Reviewer #2 (Remarks to the Author):

The authors have appropriately addressed all the comments and this reviewer does not have any further comments to make. It will be an important addition to the literature and once again this reviewer would like to thank and congratulate the authors for their work.

-Thank you very much.

Reviewer #3 (Remarks to the Author):

In this revised version of original manuscript, authors addressed many of the critiques raised and concerns noted. There is still one point that is unresolved, which is related to the reason why the AC iHLCs appear to be intrinsically impaired in their OCR, including the oxygen consumption directed to synthesis of ATP as shown in Fig 4. The attempt of the authors to explain this in the Suppl Figure 8 is clearly insufficient as the activities of complex I, II and IV do not help in elucidating the phenotype of the mitochondrial function of AC iHLCs. Although the activities of these individual complexes measured do not change, mitochondrial complexes are organized in supercomplexes structures responsible for cellular respiration, which may be altered in the AC iHLCs. This is what was indicated in the first round of revision and was not explored in the revised version.

-We truly appreciate your valuable comment. We apologize for our misinterpretation of this point that you raised earlier. We are removing Suppl Figure 8A-C as it does not help in elucidating the phenotype of the mitochondrial function that was your concern. We are intensely focused on the assembly aspect of mitochondria and expedited our experiment to address your point. The best method to address this issue is to run a native gel and observe a difference in mobility. We did so and observed a different pattern of mobility of mitochondrial supercomplexes between those from healthy and AC patient-derived iHLCs. Thank you for your valuable suggestion. We are further considering following up in this area with more advance tools in the future. Supplemental Fig 8A-C has now replaced with a new figure with native PAGE gels of mitochondrial supercomplexes. We sincerely hope this new data addresses the concern raised.

REVIEWERS' COMMENTS

Reviewer #3 (Remarks to the Author):

The authors addressed in this new version the previous concern raised regarding the assembly of supercomplexes. They acknowledge that the findings are preliminary and that further studies will be required.